# Out-of-Distribution Detection with a Single Unconditional Diffusion Model

**Alvin Heng[1], Alexandre H. Thiery[2], Harold Soh[1,3]**
[1]Department of Computer Science, National University of Singapore
[2]Department of Statistics and Data Science, National University of Singapore
[3]Smart Systems Institute, National University of Singapore
{alvinh, harold}@comp.nus.edu.sg

## Abstract

Out-of-distribution (OOD) detection is a critical task in machine learning that seeks to identify abnormal samples. Traditionally, unsupervised methods utilize a deep generative model for OOD detection. However, such approaches require a new model to be trained for each inlier dataset. This paper explores whether a single model can perform OOD detection across diverse tasks. To that end, we introduce Diffusion Paths (DiffPath), which uses a single diffusion model originally trained to perform unconditional generation for OOD detection. We introduce a novel technique of measuring the rate-of-change and curvature of the diffusion paths connecting samples to the standard normal. Extensive experiments show that with a single model, DiffPath is competitive with prior work using individual models on a variety of OOD tasks involving different distributions. Our code is publicly available at https://github.com/clear-nus/diffpath.

## 1   Introduction

Out-of-distribution (OOD) detection, also known as anomaly or outlier detection, seeks to detect abnormal samples that are far from a given distribution. This is a vital problem as deep neural networks are known to be overconfident when making incorrect predictions on outlier samples [1, 2], leading to potential issues in safety-critical applications such as robotics, healthcare, finance, and criminal justice [3]. Traditionally, OOD detection using only unlabeled data relies on training a generative model on in-distribution data. Thereafter, measures such as model likelihood or its variants are used as an OOD detection score [4–6]. An alternative approach is to utilize the excellent sampling capabilities of diffusion models (DMs) to reconstruct corrupted samples, and use the reconstruction loss as an OOD measure [7–9].

However, these conventional methods require separate generative models tailored to specific inlier distributions and require retraining if the inlier data changes, such as in continual learning setups. This prompts the question: can OOD detection be performed using a *single* generative model? We answer in the affirmative and present Diffusion Paths (DiffPath) in this paper. While the use of a single model for OOD detection has been proposed in the discriminative setting [10], to the best of our knowledge, we are the first to explore this for generative models. We believe that the generative setting is particularly salient in light of recent trends where single generative foundation models are utilized across various tasks [11, 12].

Our method utilizes a single pretrained DM. In a departure from prior works that utilize variants of likelihoods [6, 5, 4] or reconstruction losses [7–9], we propose to perform OOD detection by measuring characteristics of the forward diffusion trajectory, specifically its *rate-of-change* and *curvature*, which can be computed from the score predicted by the diffusion model. We provide theoretical and empirical analyses that motivate these

38th Conference on Neural Information Processing Systems (NeurIPS 2024).

quantities as useful OOD detectors; their magnitudes are similar for samples from the same distribution and different otherwise. We summarize our contributions as follows:

1. We introduce a novel approach to OOD detection by examining the rate-of-change and curvature along the diffusion path connecting different distributions to standard normal.

2. Through comprehensive experiments with various datasets, we show that a single generative model is competitive with baselines that necessitates separate models for each distribution.

3. We offer a theoretical framework demonstrating that our method characterizes properties of the optimal transport (OT) path between the data distribution and the standard normal.

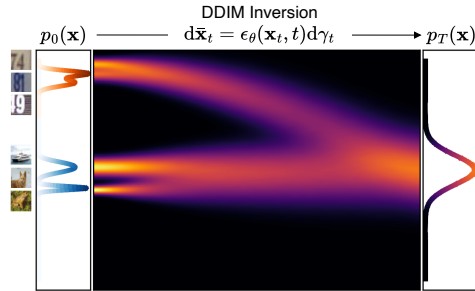

Figure 1: Illustration of the diffusion paths of samples from two different distributions (CIFAR10 and SVHN) obtained via DDIM integration. The paths have different first and second derivatives (rate-of-change and curvature). We propose to measure these quantities for OOD detection.

## 2 Background

**Score-based Diffusion Models.** Let $p_0(\mathbf{x})$ denote the data distribution. We define a stochastic differential equation (SDE), also known as the forward process, to diffuse $p_0(\mathbf{x})$ to a noise distribution $p_T(\mathbf{x})$:

$$\mathrm{d}\mathbf{x}_t = \boldsymbol{f}(\mathbf{x}_t, t)\mathrm{d}t + g(t)\mathrm{d}\mathbf{w}_t, \quad \mathbf{x}_0 \sim p_0(\mathbf{x}) \tag{1}$$

where $\boldsymbol{f}(\cdot, t) : \mathbb{R}^D \to \mathbb{R}^D$ is the drift coefficient, $g(t) \in \mathbb{R}$ is the diffusion coefficient and $\mathbf{w}_t \in \mathbb{R}^D$ is the standard Wiener process (Brownian motion). We denote $p_t$ as the marginal distribution of Eq. 1 at time $t$. By starting from noise samples $\mathbf{x}_T \sim p_T$, new samples $\mathbf{x}_0 \sim p_0(\mathbf{x})$ can be sampled by simulating the reverse SDE

$$\mathrm{d}\mathbf{x}_t = [\boldsymbol{f}(\mathbf{x}_t, t) - g(t)^2 \nabla_\mathbf{x} \log p_t(\mathbf{x}_t)]\mathrm{d}t + g(t)\mathrm{d}\bar{\mathbf{w}}_t, \quad \mathbf{x}_T \sim p_T(\mathbf{x}) \tag{2}$$

where $\bar{\mathbf{w}}_t$ is the standard Wiener process when time flows backwards from $T$ to 0, and $\mathrm{d}t$ is an infinitesimal negative timestep. The diffusion process described by Eq. 1 also has an equivalent ODE formulation, termed the probability flow (PF) ODE [13], given by

$$\mathrm{d}\mathbf{x}_t = \left[ \boldsymbol{f}(\mathbf{x}_t, t) - \frac{1}{2} g(t)^2 \nabla_\mathbf{x} \log p_t(\mathbf{x}_t) \right] \mathrm{d}t. \tag{3}$$

The ODE and SDE formulations are equivalent in the sense that trajectories under both processes share the same marginal distribution $p_t(\mathbf{x}_t)$. Hence, given an estimate of the score function $s_\theta(\mathbf{x}_t, t) \approx \nabla_\mathbf{x} \log p_t(\mathbf{x}_t)$, which can be obtained using score-matching approaches [14, 13], one can sample from the diffusion model by solving the reverse SDE or integrating the PF ODE backwards in time.

In this work, we focus on the variance-preserving formulation used in DDPM [15], which is given by an Ornstein-Uhlenbeck forward process

$$\mathrm{d}\mathbf{x}_t = -\frac{1}{2} \beta_t \mathbf{x}_t \mathrm{d}t + \sqrt{\beta_t}\mathrm{d}\mathbf{w}_t, \quad \mathbf{x}_0 \sim p_0(\mathbf{x}) \tag{4}$$

where $\beta_t$ are time-dependent constants. Under Eq. 4, diffused samples $\mathbf{x}_t$ can be sampled analytically via $p_t(\mathbf{x}_t|\mathbf{x}_0) = \mathcal{N}(\mathbf{x}_t; \sqrt{\bar{\alpha}_t}\mathbf{x}_0, \sigma_t^2 \mathbf{I})$, where $\beta_t = -\frac{\mathrm{d}}{\mathrm{d}t} \log \bar{\alpha}_t$ and $\sigma_t^2 = 1 - \bar{\alpha}_t$. The score estimator, $\boldsymbol{\epsilon}_\theta(\mathbf{x}_t, t) \approx -\sigma_t \nabla_\mathbf{x} \log p_t(\mathbf{x}_t)$, can be trained via the following objective

$$\min_\theta \mathbb{E}_{t \sim \mathcal{U}[0,1] \mathbf{x}_0 \sim p_0(\mathbf{x}_0) \mathbf{x}_t \sim p_t(\mathbf{x}_t|\mathbf{x}_0)} \left[ \|\boldsymbol{\epsilon}_\theta(\mathbf{x}_t, t) - \boldsymbol{\epsilon}\|_2^2 \right], \tag{5}$$

where $\boldsymbol{\epsilon} = -\sigma_t \nabla_\mathbf{x} \log p_t(\mathbf{x}_t|\mathbf{x}_0) = (\mathbf{x}_t - \sqrt{\bar{\alpha}_t}\mathbf{x}_0)/\sigma_t$.

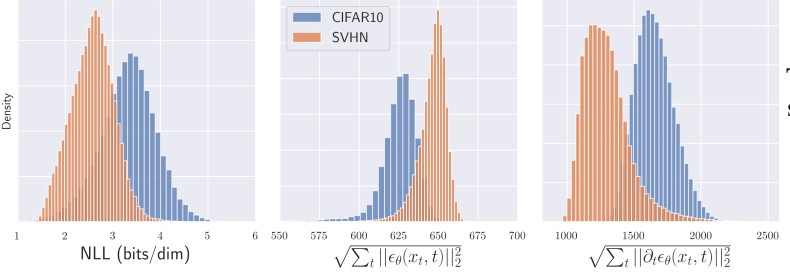

Figure 2: Histograms of various statistics of the respective training sets. The NLL is calculated using a diffusion model trained on CIFAR10, while the other two statistics are calculated with a model trained on ImageNet.

Table 1: AUROC of statistics shown in Fig. 2.

| Method | C10 *vs* SVHN |
|---|---|
| NLL | 0.091 |
| $\sqrt{\sum_t \|\boldsymbol{\epsilon}_\theta(\mathbf{x}_t, t)\|_2^2}$ | 0.856 |
| $\sqrt{\sum_t \|\partial_t \boldsymbol{\epsilon}_\theta(\mathbf{x}_t, t)\|_2^2}$ | 0.965 |

**Unsupervised OOD Detection.** Given a distribution of interest $p(\mathbf{x})$, the goal of OOD detection is to construct a scoring function which outputs a quantity $S_\theta(\mathbf{x}) \in \mathbb{R}$ that identifies if a given test point $\mathbf{x}_{\text{test}}$ is from $p(\mathbf{x})$. In this work, a higher value of $S_\theta(\mathbf{x}_{\text{test}})$ indicates that the sample is more likely to be drawn from $p(\mathbf{x})$. We will use the notation "A *vs* B" to denote the task of distinguishing samples between A and B, where A is the inlier distribution and B is the outlier distribution. In unsupervised OOD detection, one must construct the function $S_\theta$ using only knowledge of A.

## 3 Diffusion Models for OOD Detection

An overview of our method, DiffPath, is illustrated in Fig. 1. DiffPath is based on the insight that the *rate-of-change* and *curvature* of the diffusion path connecting samples to standard normal differ between distributions, making them effective indicators for OOD detection. This section outlines the methodology behind DiffPath. We begin in Sec. 3.1, where we provide evidence that likelihoods from a diffusion model are insufficient for OOD detection. Next, Sec. 3.2 shows that the score function is a measure of the rate-of-change and motivates the use of a single generative model. We then motivate the curvature as the derivative of the score in Sec. 3.3. We consider the curvature statistic as one variation of our method and abbreviate it as DiffPath-1D. In Sec. 3.4, we contextualize our method in terms of the optimal transport path between samples and standard normal, and finally propose a higher-order, hybrid variation called DiffPath-6D in Sec. 3.5, which incorporates both the rate-of-change and curvature quantities.

### 3.1 Diffusion Model Likelihoods Do Not Work for OOD Detection

When leveraging a likelihood-based generative model for OOD detection, the most natural statistic to consider is the likelihood itself. As DMs are trained to maximize the evidence lower bound (ELBO) of the data, one would expect that in-distribution samples have higher ELBO under the model compared to out-of-distribution samples. However, prior works [2, 16] have shown that the opposite behavior was observed in deep generative models, such as normalizing flows, where the model assigned higher likelihoods to OOD samples.

In Fig. 2, we plot the distributions of the negative ELBO (denoted NLL) of the CIFAR10 and SVHN training sets for a DM trained on CIFAR10. Our results corroborate earlier findings that likelihoods are not good OOD detectors; the NLL of CIFAR10 samples are higher than SVHN samples, meaning in-distribution samples have lower likelihoods than out-of-distribution samples. The poor AUROC score in Table 1 quantitatively demonstrates the inability of likelihoods to distinguish between inlier and outlier samples. This motivates us to search for better statistics that we can extract from DMs for OOD detection.

### 3.2 Scores as an OOD Statistic

**Scores as KL Divergence Proxy.** We start by rewriting the PF ODE, Eq. 3, in the following form:

$$\frac{\mathrm{d}\mathbf{x}_t}{\mathrm{d}t} = \boldsymbol{f}(\mathbf{x}_t, t) + \frac{g(t)^2}{2\sigma_t}\boldsymbol{\epsilon}_p(\mathbf{x}_t, t) \tag{6}$$

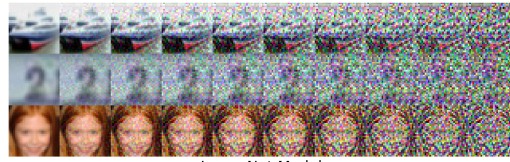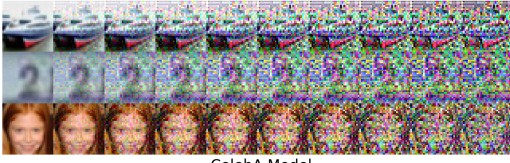

ImageNet Model                                    CelebA Model

Figure 3: Illustration of the forward integration of Eq. 7 on samples from CIFAR10, SVHN and CelebA. Both the ImageNet and CelebA models are able bring the samples approximately to standard normal. Other than the case where the CelebA model is used to integrate CelebA samples (last row of the right figure), the samples shown here have not been seen by the models during training. While in certain cases the end result appears to contain features of the original image, thus deviating from an isotropic Gaussian (e.g., first row of the right figure), empirically we find that the scores remain accurate enough for outlier detection; see Sec. 5 for quantitative results.

where we have parameterized the score as $\boldsymbol{\epsilon}_p(\mathbf{x}_t, t) = -\sigma_t \nabla_{\mathbf{x}} \log p_t(\mathbf{x})$.

**Theorem 1.** *Denote $\phi_t$ and $\psi_t$ as the marginals from evolving two distinct distributions $\phi_0$ and $\psi_0$ via their respective probability flow ODEs (Eq. 6) forward in time. We consider the case with the same forward process, i.e., the two PF ODEs have the same $\boldsymbol{f}(\mathbf{x}_t, t), g(t)$ and $\sigma_t$. Under some regularity conditions stated in Appendix A.1,*

$$D_{\mathrm{KL}}(\phi_0 \| \psi_0) = \frac{1}{2} \int_0^T \mathbb{E}_{\mathbf{x} \sim \phi_t} \frac{g(t)^2}{\sigma_t} \| \boldsymbol{\epsilon}_\phi(\mathbf{x}_t, t) - \boldsymbol{\epsilon}_\psi(\mathbf{x}_t, t) \|_2^2 \, \mathrm{d}t + D_{\mathrm{KL}}(\phi_T \| \psi_T).$$

The term $D_{\mathrm{KL}}(\phi_T \| \psi_T)$ vanishes as $\phi_T = \psi_T = \mathcal{N}(\mathbf{0}, \mathbf{I})$ by construction, assuming the true scores are available. In practice, we rely on a score estimator $\boldsymbol{\epsilon}_\theta$ obtained via score matching approaches. Theorem 1 suggests that the scores of the marginal distributions along the ODE path serve as a proxy for the KL divergence: as $D_{\mathrm{KL}}(\phi_0 \| \psi_0)$ increases, so should the difference in the norms of their scores. Another interpretation is that this difference, $\mathbb{E}[\| \boldsymbol{\epsilon}_\phi(\mathbf{x}_t, t) - \boldsymbol{\epsilon}_\psi(\mathbf{x}_t, t) \|_2^2]$, is a measure of the Fisher divergence between the two distributions, which forms the foundation for score matching [17]. This motivates using the norm of the scores as a statistic for distinguishing two distributions.

However, Theorem 1 is not immediately useful as it requires a priori knowledge of both distributions, whereas in unsupervised OOD detection, only knowledge of the inlier distribution is available. Interestingly, we empirically observe that it is possible to approximate the forward probability flow ODE for different distributions using a *single* diffusion model. Recall that as the PF ODE has the same marginal as the forward SDE, if the score estimate $\boldsymbol{\epsilon}_\theta$ has converged to the true score, then forward integration of a sample $\mathbf{x}_0$ using Eq. 6 should bring the sample to approximately standard normal, $\mathbf{x}_T \sim \mathcal{N}(\mathbf{0}, \mathbf{I})$.

Specifically, we consider the following parameterization [18] of the PF ODE

$$\mathrm{d}\bar{\mathbf{x}}_t = \boldsymbol{\epsilon}_\theta(\mathbf{x}_t, t) \mathrm{d}\gamma_t \tag{7}$$

where $\gamma_t = \sqrt{\frac{1 - \bar{\alpha}_t^2}{\bar{\alpha}_t^2}}$ and $\bar{\mathbf{x}}_t = \mathbf{x}_t \sqrt{1 + \gamma_t^2}$. Let $\boldsymbol{\epsilon}_\theta(\mathbf{x}_t, t) = -\sigma_t \nabla_{\mathbf{x}} \log p_t(\mathbf{x})$ be a score model trained on $p_0(\mathbf{x})$. It is known that the DDIM sampler [19] is Euler's method applied to Eq. 7. In Fig. 3, we integrate Eq. 7 forward in time using DDIM for samples from various distributions, most of which are unseen by the model during training. Qualitatively, we observe the surprising fact that both the ImageNet and CelebA models are able to bring the samples approximately to the standard normal. We ablate the choice of $p_0$ in Sec. 5.2.

This motivates $\boldsymbol{\epsilon}_\theta$ as a replacement for arbitrary $\boldsymbol{\epsilon}_\phi$ when integrating Eq. 7 forward with samples from $\phi_0$. In Fig. 2, we see that the distributions of $\sqrt{\sum_t \| \boldsymbol{\epsilon}_\theta(\mathbf{x}_t, t) \|_2^2}$, the square root of the sum of $L^2$ norms of scores over time, applied to the two datasets using a single model trained on ImageNet are better separated than the likelihoods. Note that Theorem 1 tells us only that the score norms of inlier and outlier samples are *different*, not whether one is higher or lower than the other. Thus, we propose the following OOD detection scheme: fit a Kernel Density Estimator (KDE) to $\sqrt{\sum_t \| \boldsymbol{\epsilon}_\theta(\mathbf{x}_t, t) \|_2^2}$ of the training set for a given distribution, then use the KDE likelihoods of a test sample as the OOD score

**Algorithm 1** OOD detection with DiffPath

---
**Input:** Trained DM $\boldsymbol{\epsilon}_\theta$, ID train set $\mathbf{X}_{\text{train}}$, test samples $\mathbf{X}_{\text{test}}$, empty lists $L_{\text{train}}$ and $L_{\text{test}}$
**Output:** OOD scores of test samples $S_\theta(\mathbf{X}_{\text{test}})$
1: **for** $\mathbf{x}_0$ in $\mathbf{X}_{\text{train}}$ **do**
2:      $\{\boldsymbol{\epsilon}_\theta(\mathbf{x}_t, t)\}_{t=0}^T \leftarrow \texttt{DDIMInversion}(\mathbf{x}_0, \boldsymbol{\epsilon}_\theta)$               ▷ Integrate Eq. 7 from $t = 0$ to $T$
3:      Calculate OOD statistic using $\{\boldsymbol{\epsilon}_\theta(\mathbf{x}_t, t)\}_{t=0}^T$
4:      Append statistic to $L_{\text{train}}$
5: **end for**
6: $p_{\text{train}}(\cdot) \leftarrow$ fit density estimate to $L_{\text{train}}$                         ▷ e.g., KDE, GMM
7: $L_{\text{test}} \leftarrow$ Repeat lines $1 - 5$ with $\mathbf{X}_{\text{test}}$
8: **return** $p_{\text{train}}(l)$ for every $l$ in $L_{\text{test}}$

---

$S_\theta$. This way, the likelihoods of outlier samples under the KDE are *lower* than for inlier samples, allowing us to compute the AUROC. We provide pseudocode in Algorithm 1. With this scheme, the AUROC scores in Table 1 show a large improvement over likelihoods; however, we would like to pursue even better OOD statistics, which we discuss in Sec. 3.3.

**Score as First-Order Taylor Expansion.** We provide a second interpretation of the score, and subsequently motivate a new statistic that can be used for OOD detection. Recall that the numerical DDIM solver is the first-order Euler's method applied to Eq. 7. In general, we can expand the ODE to higher-order terms using the truncated Taylor method [20, 18]:

$$
\begin{aligned}
\bar{\mathbf{x}}_{t_{n+1}} &= \bar{\mathbf{x}}_{t_n} + h_n \frac{d\bar{\mathbf{x}}_t}{d\gamma_t}\Big|_{(\bar{\mathbf{x}}_{t_n}, t_n)} + \frac{1}{2!} h_n^2 \frac{d^2\bar{\mathbf{x}}_t}{d\gamma_t^2}\Big|_{(\bar{\mathbf{x}}_{t_n}, t_n)} + \dots \\
&= \bar{\mathbf{x}}_{t_n} + h_n \boldsymbol{\epsilon}_\theta(\mathbf{x}_{t_n}, t_n) + \frac{1}{2!} h_n^2 \frac{d\boldsymbol{\epsilon}_\theta}{d\gamma_t}\Big|_{(\bar{\mathbf{x}}_{t_n}, t_n)} + \dots
\end{aligned}
\tag{8}
$$

where $h_n = \gamma_{t_{n+1}} - \gamma_{t_n}$. The norm of the first-order score, $\|\boldsymbol{\epsilon}_\theta\|_2$, therefore measures the *rate-of-change* of the ODE integration path. Intuitively, the ODE integration path necessary to bring different distributions to the standard normal in finite time differs (c.f. the PF ODE path is also the optimal transport path, see Sec. 3.4) , hence the rate-of-change differs as well.

### 3.3 Second-Order Taylor Expansion (DiffPath-1D)

Based on the preceding discussion, it is natural to consider if higher-order terms in the ODE Taylor expansion can also serve as an effective OOD statistic. We answer in the affirmative by considering the second order term, $\frac{d\boldsymbol{\epsilon}_\theta}{d\gamma_t}$. Intuitively, the second-order term measures the *curvature* of the ODE integration path. We expand the derivative as follows [18]:

$$
\begin{aligned}
\frac{d\boldsymbol{\epsilon}_\theta}{d\gamma_t} &= \frac{\partial\boldsymbol{\epsilon}_\theta(\mathbf{x}_t, t)}{\partial\mathbf{x}_t}\frac{d\mathbf{x}_t}{d\gamma_t} + \frac{\partial\boldsymbol{\epsilon}_\theta(\mathbf{x}_t, t)}{\partial t}\frac{dt}{d\gamma_t} \\
&= \frac{1}{\sqrt{\gamma_t^2 + 1}} \underbrace{\frac{\partial\boldsymbol{\epsilon}_\theta(\mathbf{x}_t, t)}{\partial\mathbf{x}_t}\boldsymbol{\epsilon}_\theta(\mathbf{x}_t, t)}_{\text{JVP}} - \frac{\gamma_t}{1 + \gamma_t^2} \underbrace{\frac{\partial\boldsymbol{\epsilon}_\theta(\mathbf{x}_t, t)}{\partial\mathbf{x}_t}\mathbf{x}_t}_{\text{JVP}} + \frac{\partial\boldsymbol{\epsilon}_\theta(\mathbf{x}_t, t)}{\partial t}\frac{dt}{d\gamma_t}.
\end{aligned}
\tag{9}
$$

We see that the derivative contains two Jacobian-Vector Products (JVP) and a simple time derivative term. In principle, all three terms can be computed using automatic differentiation. However, this makes inference twice as costly due to the need for an additional backward pass after every forward pass of the network, and significantly more memory-intensive due to storage of the full computation graph. Fortunately, the time derivative term can be computed using simple finite difference:

$$
\frac{\partial\boldsymbol{\epsilon}_\theta(\mathbf{x}_t, t)}{\partial t} \approx \frac{\boldsymbol{\epsilon}_\theta(\mathbf{x}_{t+\Delta t}, t + \Delta t) - \boldsymbol{\epsilon}_\theta(\mathbf{x}_t, t)}{\Delta t}
\tag{10}
$$

where the pairs $(\mathbf{x}_t, \mathbf{x}_{t+\Delta t})$ are obtained during standard DDIM integration. Thus, no additional costs associated with gradient computation are incurred.

Surprisingly, we observe that for high-dimensional distributions such as images that we consider in this work, the time derivative in Eq. 10 alone provides an accurate enough estimate for OOD

detection. Using the same ImageNet model as in Sec. 3.2, we observe an improvement in AUROC scores in CIFAR10 *vs* SVHN in Table 1 when using the second-order statistic. Qualitatively, the distributions of $\sqrt{\sum_t \|\partial_t \epsilon_\theta(\mathbf{x}_t, t)\|_2^2}$ are more spread out in Fig. 2 as compared to the first-order scores, although the distinction is subtle; the quantitative results provide a more reliable confirmation of the benefit of using the second-order statistic.

We thus consider the second-order statistic alone, $\sqrt{\sum_t \|\partial_t \epsilon_\theta(\mathbf{x}_t, t)\|_2^2}$, as a possible statistic for OOD detection. As it is a one-dimensional quantity, we abbreviate it as DiffPath-1D. We evaluate DiffPath-1D in Sec. 5.2.

### 3.4 Connections to Optimal Transport

Recent works have viewed DDIM integration as an encoder that maps the data distribution to standard normal [21, 22]. They prove that this map is the optimal transport (OT) path if the data distribution is Gaussian, while providing numerical results suggesting likewise for high-dimensional data like images. As a result, we can view the OOD statistics proposed in Sec. 3.2 and Sec. 3.3 as characterizing different derivatives of the OT path: the score $\|\epsilon_\theta\|_2$ represents the rate-of-change of the path, while the time derivative $\|\partial_t \epsilon_\theta\|_2$ represents its curvature.

To justify our proposition that derivatives of OT paths serve as meaningful OOD statistics, we consider the following toy example [21] of distinguishing two multivariate Gaussians (detailed derivation in Appendix A.2). Let the distributions be $p_0^i(\mathbf{x}) \sim \mathcal{N}(\mathbf{a}_i, \mathbf{I}), i \in \{0, 1\}$, where $\mathbf{a}_i \in \mathbb{R}^d$ and $\mathbf{I} \in \mathbb{R}^{d \times d}$. The marginal densities along the forward diffusion can be computed exactly using the transition formulas for SDEs [23] and is given by $p_t^i(\mathbf{x}) \sim \mathcal{N}(\mathbf{a}_i e^{-t}, \mathbf{I})$, with PF ODE $\frac{d\mathbf{x}_i}{dt} = -\mathbf{a}_i e^{-t}$. This path corresponds exactly to the OT map between $p_0^i$ and $\mathcal{N}(\mathbf{0}, \mathbf{I})$. In this case, the corresponding first and second-order OOD statistics are equal and given by $\left\|\frac{d\mathbf{x}_i}{dt}\right\|_2 = \left\|\frac{d^2\mathbf{x}_i}{dt^2}\right\|_2 = \|\mathbf{a}_i e^{-t}\|_2$. Crucially, they are proportional to $\|\mathbf{a}_i\|_2$, meaning that as the two distributions move farther apart (i.e., as $\|\mathbf{a}_0 - \mathbf{a}_1\|_2$ increases), so should the $L^2$ norms of the OOD statistics, thereby increasing their ability to distinguish samples between the two.

### 3.5 Higher-dimensional Statistic (DiffPath-6D)

Owing to its simplicity, the one-dimensional statistic proposed in Sec. 3.3 may suffer from edge cases or perform suboptimally as information is condensed to a single scalar. For instance, given an image $\mathbf{x}_0$ with pixels normalized to the range $[-1, 1]$, one such edge case is distinguishing $\mathbf{x}_0$ from itself with the sign of the pixels flipped, $-\mathbf{x}_0$. The two samples will produce symmetric OT paths differing only by a negative sign, resulting in identical statistics after taking the $L^2$ norm. We can see this from Table 2 where DiffPath-1D fails to distinguish CIFAR10 samples from itself with signs flipped, which we call negative CIFAR10. As such, we propose a higher-dimensional statistic that

Table 2: AUROC of DiffPath 1D *vs* 6D.

| Method | C10 *vs* neg. C10 |
|---|---|
| 1D | 0.500 |
| 6D | 0.994 |

does not utilize the standard form of the $L^p$ norm: $\|\mathbf{x}\|_p = \sum_i |\mathbf{x}_i|^p$. We define a new scalar quantity, $\langle \mathbf{x} \rangle_p = \sum_i (\mathbf{x}_i)^p$, which retains the sign information, and propose a new six-dimensional statistic we dub DiffPath-6D:

$$\left[ \sum_t \langle \epsilon_\theta(\mathbf{x}_t, t) \rangle_1, \sum_t \langle \epsilon_\theta(\mathbf{x}_t, t) \rangle_2, \sum_t \langle \epsilon_\theta(\mathbf{x}_t, t) \rangle_3, \sum_t \langle \partial_t \epsilon_\theta(\mathbf{x}_t, t) \rangle_1, \sum_t \langle \partial_t \epsilon_\theta(\mathbf{x}_t, t) \rangle_2, \sum_t \langle \partial_t \epsilon_\theta(\mathbf{x}_t, t) \rangle_3 \right]^\top$$

which concatenates scalars based on the first, second and third powers of $\epsilon_\theta$ and $\partial_t \epsilon_\theta$ into a vector. From Table 2, DiffPath-6D is able to distinguish CIFAR10 from neg. CIFAR10 near perfectly. We validate both DiffPath-1D and DiffPath-6D on a wider suite of experiments in Sec. 5.

## 4 Related Works

Modern OOD detection can be divided roughly into three categories: feature-based, likelihood-based, reconstruction-based. Feature-based methods extract features from inlier samples and fit a likelihood or distance function as an OOD detector. For instance, one can obtain the latent representations of a test sample using an autoencoder and measure its Mahalanobis distance to the representations of inlier samples [24]. Distances between features derived from self-supervised learning models are

also utilized in similar contexts [25, 26]. Similar to our work, Xiao et al. [10] showed that one can perform OOD detection using features from a single discriminative model.

Likelihood-based approaches leverage a generative model trained on inlier samples. These methods typically employ variants of the log-likelihood of a test sample under the model as the OOD detection score. Nalisnick et al. [2] first pointed out that deep generative models might erroneously assign higher likelihoods to outlier samples. Several explanations have been proposed, such as the input complexity [16] and typicality [27] of samples. As a result, just as we show in Sec. 3.1, vanilla likelihoods are rarely used. Instead, variants derived from the log-likelihood have been proposed, such as likelihood ratios [4], ensembles of the likelihood [5], density of states [6], energy-based models [28] and typicality tests [27]. Diffusion Time Estimation [29] estimates the distribution over the diffusion time of a noisy test sample and uses the mean or mode as the OOD score. MSMA [30] uses the score function over discrete noise levels for OOD detection. One can view MSMA as a specific case of DiffPath which only utilizes the first-order statistic, while we generalize to higher-order terms. MSMA proposes to measure the score at various noise levels, while our method sums over the entire diffusion path. It is worth emphasizing that MSMA requires different models for different inlier distributions, unlike our single model setup.

Reconstruction-based approaches evaluate how well a generative model, trained on in-distribution data, can reconstruct a test sample. Earlier works utilize autoencoders [31] and GANs [32] for reconstruction. Recent works have adapted unconditional DMs to this approach due to its impressive sample quality. A test sample is first artificially corrupted before being reconstructed using the DM's sampling process. DDPM-OOD [7] noises a sample using the forward process and evaluates the perceptual quality [33] of the reconstructed sample. Projection Regret [9] adopts a similar approach, but uses a Consistency Model [34] and introduces an additional projection regret score. LMD [8] corrupts the image by masking and reconstructs the sample via inpainting. Evidently, DiffPath differs from these diffusion approaches as we do not utilize reconstructions. We also stress again that these baselines require different models for different inlier tasks.

## 5 Experiments

Based on our earlier analysis, we hypothesize that DiffPath can be utilized for OOD detection across diverse tasks using a single model. In this section, we validate our hypothesis with comprehensive experiments across numerous pairwise OOD detection tasks and compare DiffPath's performance against state-of-the-art baselines.

**Datasets.** All experiments are conducted as of pairwise OOD detection tasks using CIFAR10 (C10), SVHN, and CelebA as inlier datasets, and CIFAR100 (C100) and Textures as additional outlier datasets. Unconditional diffusion models are employed at resolutions of $32 \times 32$ and $64 \times 64$. The model utilizing ImageNet as the base distribution is trained at $64 \times 64$ resolution, while all other models are trained at $32 \times 32$.

**Methodology and Baselines.** Our methodology features two variants of our model, DiffPath-1D using KDE and DiffPath-6D using a Gaussian Mixture Model for OOD scoring, as outlined in Sec. 3. We compare against a variety of generative baselines, including Energy-based Model (EBM) such as IGEBM [28], VAEBM [35] and Improved CD [36], as well as Input Complexity (IC) [16], Density of States (DOS) [6], Watanabe-Akaike Information Criterion (WAIC) [5], Typicality Test (TT) [27] and Likelihood Ratio (LR) [4] applied to the Glow [37] model. Additionally, we compare against diffusion baselines such as vanilla NLL and IC based on the DM's likelihoods and re-implementations of DDPM-OOD [7], LMD [8], and MSMA [30] based on open-source code for full comparisons.

### 5.1 Main Results

Table 3 summarizes our main results. Here, we report outcomes for DiffPath-6D using ImageNet and CelebA as base distributions. DiffPath-6D-CelebA achieves an average AUROC of 0.931, comparable to the leading diffusion-based approach MSMA and outperforming all other baselines, while utilizing only a single model. Similar to MSMA, we attain this result using 10 NFEs, significantly surpassing other diffusion baselines that require an order of magnitude or more NFEs. When using ImageNet as the base distribution, the average AUROC of 0.850 is competitive with LMD, which requires several

Table 3: AUROC scores for various in-distribution *vs* out-of-distribution tasks. Higher is better. DiffPath-6D-ImageNet and DiffPath-6D-CelebA denote our method using diffusion models trained with ImageNet and CelebA as base distributions respectively. **Bold** and underline denotes the best and second best result respectively. We also show the number of function evaluations (NFE) for diffusion methods, where lower is better.

| Method | C10 *vs* | | | | SVHN *vs* | | | | CelebA *vs* | | | | Average | NFE |
|---|---|---|---|---|---|---|---|---|---|---|---|---|---|---|
| | SVHN | CelebA | C100 | Textures | C10 | CelebA | C100 | Textures | C10 | SVHN | C100 | Textures | | |
| IC | 0.950 | 0.863 | 0.736 | - | - | - | - | - | - | - | - | - | - | - |
| IGEBM | 0.630 | 0.700 | 0.500 | 0.480 | - | - | - | - | - | - | - | - | - | - |
| VAEBM | 0.830 | 0.770 | 0.620 | - | - | - | - | - | - | - | - | - | - | - |
| Improved CD | 0.910 | - | **0.830** | 0.880 | - | - | - | - | - | - | - | - | - | - |
| DoS | 0.955 | 0.995 | 0.571 | - | 0.962 | **1.00** | 0.965 | - | 0.949 | 0.997 | 0.956 | - | 0.928 | - |
| WAIC[1] | 0.143 | 0.928 | 0.532 | - | 0.802 | 0.991 | 0.831 | - | 0.507 | 0.139 | 0.535 | - | 0.601 | - |
| TT[1] | 0.870 | 0.848 | 0.548 | - | 0.970 | **1.00** | 0.965 | - | 0.634 | 0.982 | 0.671 | - | 0.832 | - |
| LR[1] | 0.064 | 0.914 | 0.520 | - | 0.819 | 0.912 | 0.779 | - | 0.323 | 0.028 | 0.357 | - | 0.524 | - |
| *Diffusion-based* | | | | | | | | | | | | | | |
| NLL | 0.091 | 0.574 | 0.521 | 0.609 | **0.990** | 0.999 | **0.992** | 0.983 | 0.814 | 0.105 | 0.786 | 0.809 | 0.689 | 1000 |
| IC | 0.921 | 0.516 | 0.519 | 0.553 | 0.080 | 0.028 | 0.100 | 0.174 | 0.485 | 0.972 | 0.510 | 0.559 | 0.451 | 1000 |
| MSMA | 0.957 | **1.00** | 0.615 | **0.986** | 0.976 | 0.995 | 0.980 | **0.996** | 0.910 | 0.996 | 0.927 | **0.999** | 0.945 | **10** |
| DDPM-OOD | 0.390 | 0.659 | 0.536 | 0.598 | 0.951 | 0.986 | 0.945 | 0.910 | 0.795 | 0.636 | 0.778 | 0.773 | 0.746 | 350 |
| LMD | **0.992** | 0.557 | 0.604 | 0.667 | 0.919 | 0.890 | 0.881 | 0.914 | 0.989 | **1.00** | 0.979 | 0.972 | 0.865 | $10^4$ |
| *Ours* | | | | | | | | | | | | | | |
| DiffPath-6D-ImageNet | 0.856 | 0.502 | 0.580 | 0.841 | 0.943 | 0.964 | 0.954 | 0.969 | 0.807 | 0.981 | 0.843 | 0.964 | 0.850 | **10** |
| DiffPath-6D-CelebA | 0.910 | 0.897 | 0.590 | 0.923 | 0.939 | 0.979 | 0.953 | 0.981 | **0.998** | **1.00** | **0.998** | **0.999** | 0.931 | **10** |

Table 4: Ablation results when we vary $p_0(\mathbf{x})$, the distribution the single DM is trained on. We use DiffPath-6D with 10 NFEs. Random denotes a randomly initialized model.

| $q_0(\mathbf{x})$ | C10 *vs* | | | | SVHN *vs* | | | | CelebA *vs* | | | | Average |
|---|---|---|---|---|---|---|---|---|---|---|---|---|---|
| | SVHN | CelebA | C100 | Textures | C10 | CelebA | C100 | Textures | C10 | SVHN | C100 | Textures | |
| C10 | **0.939** | 0.484 | **0.604** | 0.870 | 0.961 | 0.961 | 0.973 | 0.982 | 0.719 | 0.997 | 0.796 | 0.950 | 0.853 |
| SVHN | 0.742 | 0.482 | 0.579 | 0.872 | **0.991** | **0.994** | **0.992** | **0.989** | 0.706 | 0.974 | 0.769 | 0.961 | 0.838 |
| CelebA | 0.910 | **0.897** | 0.590 | **0.923** | 0.939 | 0.979 | 0.953 | 0.981 | **0.998** | **1.00** | **0.998** | **0.999** | **0.931** |
| ImageNet | 0.856 | 0.502 | 0.580 | 0.841 | 0.943 | 0.964 | 0.954 | 0.969 | 0.807 | 0.981 | 0.843 | 0.964 | 0.850 |
| Random | 0.338 | 0.426 | 0.538 | 0.31 | 0.665 | 0.592 | 0.693 | 0.471 | 0.577 | 0.411 | 0.612 | 0.381 | 0.501 |

magnitudes more NFEs due to multiple reconstructions. This is despite the ImageNet model never having seen any samples from the evaluated distributions during training.

The empirical results indicate that the CelebA-based model outperforms the ImageNet-based model primarily due to its superior performance on tasks involving CelebA samples, whether they are in-distribution or out-of-distribution. For instance, DiffPath-6D-CelebA achieves near-perfect performance on all tasks where CelebA is in-distribution (rightmost columns), and in the CIFAR10 *vs* CelebA task. On tasks that do not involve CelebA samples, the two models exhibit roughly comparable performance. This suggests that distinguishing CelebA from other samples is particularly challenging, and that DiffPath benefits from exposure to inlier samples from the respective distributions during training. Next, we discuss the effect of the base datasets and other design considerations via ablations.

---

[1]Results obtained from Morningstar et al. [6].

Table 5: Ablation on the number of DDIM steps (NFE). We use DiffPath-6D-CelebA.

| | C10 vs | | | | SVHN vs | | | | CelebA vs | | | | |
|---|---|---|---|---|---|---|---|---|---|---|---|---|---|
| NFEs | SVHN | CelebA | C100 | Textures | C10 | CelebA | C100 | Textures | C10 | SVHN | C100 | Textures | Average |
| 5 | **0.916** | **0.928** | 0.584 | 0.900 | **0.955** | 0.940 | **0.960** | 0.973 | **0.999** | 1.00 | **0.998** | 0.997 | 0.929 |
| 10 | 0.910 | 0.897 | **0.590** | 0.923 | 0.939 | 0.979 | 0.953 | **0.981** | 0.998 | 1.00 | 0.998 | 0.999 | **0.931** |
| 25 | 0.898 | 0.866 | 0.578 | **0.933** | 0.882 | **0.997** | 0.906 | 0.979 | 0.996 | 1.00 | 0.995 | 0.996 | 0.919 |
| 50 | 0.896 | 0.831 | 0.575 | 0.931 | 0.853 | 0.996 | 0.879 | 0.974 | 0.991 | 1.00 | 0.991 | 0.997 | 0.910 |

Table 6: Ablation results comparing DiffPath-1D and DiffPath-6D using models trained with ImageNet and CelebA as base distributions.

| | C10 vs | | | | SVHN vs | | | | CelebA vs | | | | |
|---|---|---|---|---|---|---|---|---|---|---|---|---|---|
| Method | SVHN | CelebA | C100 | Textures | C10 | CelebA | C100 | Textures | C10 | SVHN | C100 | Textures | Average |
| DiffPath-1D-ImageNet | **0.965** | 0.394 | 0.551 | 0.685 | **0.971** | **0.986** | **0.972** | 0.949 | 0.693 | 0.991 | 0.721 | 0.797 | 0.806 |
| DiffPath-6D-ImageNet | 0.856 | 0.502 | **0.580** | 0.841 | 0.943 | 0.964 | 0.954 | 0.969 | 0.807 | 0.981 | 0.843 | 0.964 | 0.850 |
| DiffPath-1D-CelebA | 0.956 | 0.811 | 0.545 | 0.688 | 0.948 | 0.690 | 0.933 | 0.932 | 0.899 | 0.666 | 0.881 | 0.911 | 0.822 |
| DiffPath-6D-CelebA | 0.910 | **0.897** | 0.59 | **0.923** | 0.939 | 0.979 | 0.953 | **0.981** | **0.998** | **1.00** | **0.998** | **0.999** | **0.931** |

## 5.2 Ablations

**Choice of Diffusion Training Set.** We investigate the impact of the base dataset on the performance of DiffPath-6D. In Table 4, we compare four different base distributions alongside a randomly initialized model. As a baseline check, we observe that the average AUROC of the randomly initialized model is 0.501, which is consistent with random guessing. This indicates that training on a base distribution is essential for the model to learn features for effective OOD detection.

Our ablations include CIFAR10 and SVHN as base distributions, in addition to CelebA and ImageNet shown in Table 3. Among these four base distributions, CelebA yields the best performance overall. Notably, the models trained on SVHN and CelebA demonstrate superior results when the inlier data aligns with their respective training distributions. This supports the established principle of training models on in-distribution samples, and we show that DiffPath-6D similarly benefits from such training. However, we underscore the key finding of our work: while in-distribution training enhances performance, it is not strictly necessary. DiffPath-6D exhibits strong performance even on tasks involving samples from distributions that the model has not encountered during training.

**DiffPath 1D vs 6D.** Here we ablate on the choice of DiffPath-1D and DiffPath-6D. Table 6 shows that DiffPath-6D performs better than its 1D counterpart for both choices of base distributions. We attribute this to the increased robustness of aggregating multiple statistics, c.f. Sec. 3.5, and recommend practitioners to use DiffPath-6D in general. However, DiffPath-1D outperforms DiffPath-6D in certain instances. For instance, for the ImageNet model, DiffPath-1D achieves the best performance on CIFAR10 vs SVHN and in three out of four tasks where SVHN is the inlier distribution.

**Number of DDIM Steps.** We investigate how the performance of our method varies with the number of NFEs (DDIM steps) in Table 5. Overall, the changes in average AUROC are relatively minor as the NFEs are varied, suggesting that DiffPath is robust to the number of integration steps. The best result is obtained at 10 NFEs. While the finite difference approximation of the derivative (Eq. 10) should become more accurate as the number of NFEs increases, the aggregation of multiple statistics involving scores and its derivatives makes this effect less pronounced. We leave the investigation of this phenomena in greater detail to future work.

## 5.3 Proper Image Resizing with a Single Model

Using a single DM with a fixed input resolution necessitates resizing all images to match the model's resolution during evaluation. However, when datasets have differing original resolutions, naive resizing—upsampling lower-resolution images and downsampling higher-resolution ones—can lead

Table 7: Difference in performance when the incorrect resizing technique is used, which leads to overly optimistic results. Results on DiffPath-6D with ImageNet $64 \times 64$ as the base distribution. The results with asterisk (*) denote the scores that have been computed inaccurately.

| | C10 *vs* | | | | SVHN *vs* | | | | CelebA *vs* | | | | |
|---|---|---|---|---|---|---|---|---|---|---|---|---|---|
| Correct Resizing | SVHN | CelebA | C100 | Textures | C10 | CelebA | C100 | Textures | C10 | SVHN | C100 | Textures | Average |
| No | 0.856 | 0.999* | 0.580 | 0.999* | 0.943 | 1.00* | 0.954 | 1.00* | 0.998* | 1.00* | 0.998* | 0.981 | 0.942 |
| Yes | 0.856 | 0.502 | 0.580 | 0.841 | 0.943 | 0.964 | 0.954 | 0.969 | 0.807 | 0.981 | 0.843 | 0.964 | 0.850 |

to evaluation inaccuracies. Specifically, upsampling introduces blurriness due to pixel interpolation, while downsampling does not. This discrepancy allows the model to differentiate samples based on image blur rather than semantic content, resulting in overly optimistic performance metrics.

For instance, when evaluating DiffPath-6D trained at $64 \times 64$ pixels on the CelebA *vs* CIFAR10 task, CIFAR10 images are upsampled (introducing blur) while CelebA images are downsampled. This imbalance enables trivial distinction between the samples. As illustrated in the first row of Table 7, tasks where only one distribution undergoes upsampling yield artificially high AUROC scores.

To mitigate this issue, we propose equalizing the resizing process by first downsampling higher-resolution images to the lower resolution of the other distribution, then upsampling both to the model's required resolution. In the CelebA *vs* CIFAR10 example, CelebA images are downsampled to $32 \times 32$ pixels before both samples are upsampled to $64 \times 64$ pixels. This method ensures consistent blurring effects across all samples, reducing confounding factors. The second row of Table 7 demonstrates more accurate evaluations using this approach. We adopt this resizing procedure in all relevant experiments to ensure fair comparisons. In short, we highlight the importance of consistent image resizing for fair evaluation in OOD detection, which we hope will guide future research.

## 5.4 Near-OOD Tasks

Near-OOD tasks refer to setups where the inlier and outlier samples are semantically similar, making them challenging for most methods. From Table 3, DiffPath, like most baselines, does not perform strongly on near-OOD tasks like CIFAR10 *vs* CIFAR100. This motivated us to conduct further near-OOD experiments, the results of which are presented in Table 8 of Sec. C of the appendix. Note that near-OOD tasks are not a standard evaluation on generative methods. We defer detailed analysis of the results to the appendix, and leave further investigations on near-OOD tasks to future work.

## 6 Conclusion

In this work, we proposed Diffusion Paths (DiffPath), a method of OOD detection using a single diffusion model by characterizing properties of the forward diffusion path. In light of the growing popularity of generative foundation models, our work demonstrates that a single diffusion model can also be applied to OOD detection. There are several interesting future directions that arise from our work; for instance, applying DiffPath to other modalities such as video, audio, language, time series and tabular data, as well as investigating if higher-order terms of the Taylor expansion, or leveraging different instantiations of the PF ODE might lead to better performance.

**Limitations and Future Work.** We only calculate the simple time derivative and found that it works well experimentally, although one might compute the full derivative to quantify the curvature completely. We leave this for future work. Also, we consider DMs trained on natural images like CelebA and ImageNet, which may not be appropriate for specialized applications such as medical images. For such purposes, one could consider including domain-specific data during training.

## 7 Acknowledgements

This research is supported by the National Research Foundation Singapore and DSO National Laboratories under the AI Singapore Programme (AISG Award No: AISG2-RP-2020-017). AHT acknowledges support from the Singaporean Ministry of Education Grant MOE-000537-01 and MOE-000618-01. We would like to thank Pranav Goyal for help with the experiments.

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

# Supplementary Material for "Out-of-Distribution Detection with a Single Unconditional Diffusion Model"

## A Proofs

### A.1 Theorem 1

**Theorem 1.** *Denote $\phi_t$ and $\psi_t$ as the marginals from evolving two distinct distributions $\phi_0$ and $\psi_0$ via their respective probability flow ODEs (Eq. 6) forward in time. We consider the case with the same forward process, i.e., the two PF ODEs have the same $\boldsymbol{f}(\mathbf{x}_t, t), g(t)$ and $\sigma_t$. Under some regularity conditions stated in Appendix A.1,*

$$D_{\mathrm{KL}}(\phi_0\|\psi_0) = \frac{1}{2}\int_0^T \mathbb{E}_{\mathbf{x}\sim\phi_t}\frac{g(t)^2}{\sigma_t}\|\boldsymbol{\epsilon}_\phi(\mathbf{x}_t, t) - \boldsymbol{\epsilon}_\psi(\mathbf{x}_t, t)\|_2^2\, \mathrm{d}t + D_{\mathrm{KL}}(\phi_T\|\psi_T).$$

*Proof.* The proof is a modification from Song et al. [38]. Let us first state the PF ODEs of the two distributions explicitly:

$$\frac{\mathrm{d}\mathbf{x}_t}{\mathrm{d}t} = \boldsymbol{f}(\mathbf{x}_t, t) + \frac{g(t)^2}{2\sigma_t}\boldsymbol{\epsilon}_\phi(\mathbf{x}_t, t), \quad \boldsymbol{\epsilon}_\phi(\mathbf{x}_t, t) = -\sigma_t\nabla_\mathbf{x}\log\phi_t(\mathbf{x}) \tag{11}$$

$$\frac{\mathrm{d}\mathbf{x}_t}{\mathrm{d}t} = \boldsymbol{f}(\mathbf{x}_t, t) + \frac{g(t)^2}{2\sigma_t}\boldsymbol{\epsilon}_\psi(\mathbf{x}_t, t), \quad \boldsymbol{\epsilon}_\psi(\mathbf{x}_t, t) = -\sigma_t\nabla_\mathbf{x}\log\psi_t(\mathbf{x}). \tag{12}$$

We make the following assumption about $\phi_t$ and $\psi_t$:

$$\forall t \in [0, T],\ \exists k > 0 \text{ s.t. } \phi_t(\mathbf{x}) = O(e^{-\|\mathbf{x}\|_2^k}),\ \psi_t(\mathbf{x}) = O(e^{-\|\mathbf{x}\|_2^k}) \text{ as } \|\mathbf{x}\|_2 \to \infty. \tag{13}$$

We start by rewriting the KL divergence between $\phi_0$ and $\psi_0$ in integral form:

$$\begin{aligned} D_{\mathrm{KL}}(\phi_0\|\psi_0) &= D_{\mathrm{KL}}(\phi_0\|\psi_0) - D_{\mathrm{KL}}(\phi_T\|\psi_T) + D_{\mathrm{KL}}(\phi_T\|\psi_T) \\ &= -\int_0^T \frac{\partial D_{\mathrm{KL}}(\phi_t\|\psi_t)}{\partial t}\mathrm{d}t + D_{\mathrm{KL}}(\phi_T\|\psi_T). \end{aligned} \tag{14}$$

As we can treat the PF ODE as a special case of an SDE with zero diffusion term, we can obtain the Fokker-Planck of the marginal density of the PF ODEs, also known as the continuity equation, as follows:

$$\frac{\partial\phi_t}{\partial t} = \nabla_\mathbf{x}\cdot\left(-\boldsymbol{f}(\mathbf{x}_t, t)\phi_t(\mathbf{x}) - \frac{g(t)^2}{2\sigma_t}\boldsymbol{\epsilon}_\phi(\mathbf{x}_t, t)\phi_t(\mathbf{x})\right) = \nabla_\mathbf{x}\cdot(\boldsymbol{h}_\phi\phi_t(\mathbf{x})) \tag{15}$$

where we define $\boldsymbol{h}_\phi := -\boldsymbol{f}(\mathbf{x}_t, t) - \frac{g(t)^2}{2\sigma_t}\boldsymbol{\epsilon}_\phi(\mathbf{x}_t, t)$ for simplicity. Similarly, $\frac{\partial\psi_t}{\partial t} = \nabla_\mathbf{x}\cdot(\boldsymbol{h}_\psi\psi_t(\mathbf{x}))$. Let us now rewrite the time-derivative $\frac{\partial D_{\mathrm{KL}}(\phi_t\|\psi_t)}{\partial t}$ in Eq. 14 as follows:

$$\begin{aligned} \frac{\partial D_{\mathrm{KL}}(\phi_t\|\psi_t)}{\partial t} &= \frac{\partial}{\partial t}\int \phi_t(\mathbf{x})\log\frac{\phi_t(\mathbf{x})}{\psi_t(\mathbf{x})}\mathrm{d}\mathbf{x} \\ &= \int \frac{\partial\phi_t}{\partial t}\log\frac{\phi_t(\mathbf{x})}{\psi_t(\mathbf{x})}\mathrm{d}\mathbf{x} + \underbrace{\int\frac{\partial\phi_t(\mathbf{x})}{\partial t}\mathrm{d}\mathbf{x}}_{=0} - \int\frac{\phi_t(\mathbf{x})}{\psi_t(\mathbf{x})}\frac{\partial\psi_t(\mathbf{x})}{\partial t}\mathrm{d}\mathbf{x} \\ &= \int\nabla_\mathbf{x}\cdot(\boldsymbol{h}_\phi(\mathbf{x}, t)\phi_t(\mathbf{x}))\log\frac{\phi_t(\mathbf{x})}{\psi_t(\mathbf{x})}\mathrm{d}\mathbf{x} - \int\frac{\phi_t(\mathbf{x})}{\psi_t(\mathbf{x})}\nabla_\mathbf{x}\cdot(\boldsymbol{h}_\psi(\mathbf{x}, t)\psi_t(\mathbf{x}))\mathrm{d}\mathbf{x} \\ &\overset{(i)}{=} \int\phi_t(\mathbf{x})[\boldsymbol{h}_\phi^\top(\mathbf{x}, t) - \boldsymbol{h}_\psi^\top(\mathbf{x}, t)][\nabla_\mathbf{x}\log\phi_t(\mathbf{x}) - \nabla_x\log\psi_t(\mathbf{x})]\mathrm{d}\mathbf{x} \\ &= -\frac{1}{2}\int\phi_t(\mathbf{x})\frac{g(t)^2}{\sigma_t}\|\boldsymbol{\epsilon}_\phi(\mathbf{x}, t) - \boldsymbol{\epsilon}_\psi(\mathbf{x}, t)\|_2^2\,\mathrm{d}\mathbf{x}, \end{aligned}$$

where (i) is due to integration by parts and the fact that $\lim_{\mathbf{x}\to\infty}\boldsymbol{h}_\phi(\mathbf{x}, t)\phi_t(\mathbf{x}) = 0$ and $\lim_{\mathbf{x}\to\infty}\boldsymbol{h}_\psi(\mathbf{x}, t)\psi_t(\mathbf{x}) = 0$ due to assumption 13. Combining with Eq. 14 gives us the desired result:

$$D_{\mathrm{KL}}(\phi_0\|\psi_0) = \frac{1}{2}\int_0^T \mathbb{E}_{x\sim\phi_t}\frac{g(t)^2}{\sigma_t}\|\boldsymbol{\epsilon}_\phi(\mathbf{x}_t, t) - \boldsymbol{\epsilon}_\psi(\mathbf{x}_t, t)\|_2^2\, \mathrm{d}t + D_{\mathrm{KL}}(\phi_T\|\psi_T). \tag{16}$$

$\square$

## A.2 OT Toy Example

We derive here in detail the toy example discussed in Sec. 3.4, where we will prove the optimal transport map between a multivariate Gaussian and standard normal is identical to the diffusion PF ODE path. We consider our source distribution as $p_0(\mathbf{x}) \sim \mathcal{N}(\mathbf{a}, \mathbf{I})$, $\mathbf{a} \in \mathbb{R}^d$ and $\mathbf{I} \in \mathbb{R}^{d \times d}$. We choose our forward SDE to be parameterized as:

$$\mathrm{d}\mathbf{x}_t = -\mathbf{x}\mathrm{d}t + \sqrt{2}\mathrm{d}\mathbf{w}_t, \tag{17}$$

which is the same Ornstein–Uhlenbeck process as the DDPM forward SDE Eq. 4 with a constant noise schedule $\beta(t) = 2$. This is also commonly referred to as the Langevin equation.

As Eq. 17 has affine drift coefficients and a starting distribution which is normal, we know that the marginal distributions at intermediate times are also normal, $p_t(\mathbf{x}) \sim \mathcal{N}(\boldsymbol{\mu}(t), \boldsymbol{\Sigma}(t))$. Furthermore, we can calculate the means and variances analytically using Eq. 5.50 and Eq. 5.51 of Särkkä and Solin [23]:

$$\frac{\mathrm{d}\boldsymbol{\mu}(t)}{\mathrm{d}t} = -\boldsymbol{\mu}(t), \quad \frac{\mathrm{d}\boldsymbol{\Sigma}(t)}{\mathrm{d}t} = -2\boldsymbol{\Sigma}(t) + 2 \tag{18}$$

with solutions

$$\boldsymbol{\mu}(t) = \boldsymbol{\mu}(0)e^{-t} = \mathbf{a}e^{-t}, \quad \boldsymbol{\Sigma}(t) = \mathbf{I} + e^{-2t}(\boldsymbol{\Sigma}(0) - \mathbf{I}) = \mathbf{I}. \tag{19}$$

Thus, the marginal density has the form $p_t(\mathbf{x}) = \mathcal{N}(\mathbf{a}e^{-t}, \mathbf{I})$, from which we compute the score as $\nabla_{\mathbf{x}} \log p_t(\mathbf{x}) = -\mathbf{x} + \mathbf{a}e^{-t}$. We can substitute this into the corresponding PF ODE to obtain:

$$\begin{aligned} \frac{\mathrm{d}\mathbf{x}_t}{\mathrm{d}t} &= -\mathbf{x} - \nabla_{\mathbf{x}} \log p_t(\mathbf{x}) \\ &= -\mathbf{a}e^{-t}. \end{aligned} \tag{20}$$

The optimal transport map denoted $E_{p_0}(\mathbf{x})$ is obtained by solving Eq. 20 to get $\mathbf{x}_t = \mathbf{x}_0 + \mathbf{a}(e^{-t} - \mathbf{I})$ and taking the limit $t \to \infty$. This gives us $E_{p_0}(\mathbf{x}) = \mathbf{x} - \mathbf{a}$, which is precisely the OT map between $p_0(\mathbf{x}) \sim \mathcal{N}(\mathbf{a}, \mathbf{I})$ and $\mathcal{N}(\mathbf{0}, \mathbf{I})$ (cf. Eq. 2.40 in Peyré and Cuturi [39]).

## B  Experimental Details

**DiffPath.**  As mentioned in Sec. 5, we utilize a single unconditional diffusion model trained on CelebA and ImageNet at $32 \times 32$ and $64 \times 64$ resolution respectively. We train our own CelebA model and utilize the ImageNet checkpoint trained using Improved DDPM's $L_{\text{hybrid}}$ objective (Eq. 16 of Nichol and Dhariwal [40]) from the official repository[2]. Both models use a cosine noise schedule with a total of 4000 diffusion steps. For DiffPath-1D, we fit a KDE using a Gaussian kernel with a bandwith of 5. For DiffPath-6D, we fit a GMM with hyperparameters obtained by sweeping over a predefined number of mixture components (e.g., 50, 100) and covariance type (e.g., diagonal, full, tied). Both are implemented using the `sklearn` library.

On a single Nvidia A5000 GPU, DiffPath takes approximately 0.25s and 0.94s per integration step on $32 \times 32$ and $64 \times 64$ images respectively with a batch size of 256.

**Diffusion Baselines.**  For all diffusion baselines, we rely on the official GitHub repositories and open-source checkpoints where possible. The repositories are listed as follows: MSMA[3], DDPM-OOD[4], LMD[5]. For NLL and IC, we use the pre-trained CIFAR10 checkpoint from Improved DDPM and train our own model for SVHN using the same hyperparameters at $32 \times 32$ resolution. We calculate the NLL using the default implementation in Improved DDPM, while we compute IC using code from the LR repository[6] due to lack of official code from the IC authors. We train all baselines using 1-3 A5000 GPUs.

Table 8: Average AUROC results for near-OOD tasks as proposed in OpenOOD [41]. We use DiffPath-6D with ImageNet as the base distribution with 10 DDIM steps. **Bold** and underline denotes the best and second best result respectively. We also show the number of function evaluations (NFE) for diffusion methods, where lower is better.

| Method | CIFAR10 | TinyImageNet |
|---|---|---|
| KLM | 0.792 | 0.808 |
| VIM | 0.887 | 0.787 |
| KNN | **0.907** | 0.816 |
| DICE | 0.783 | 0.818 |
| DiffPath-6D | 0.607 | **0.845** |

## C   Near-OOD Results

To further investigate the performance of DiffPath on near-OOD tasks, we ran experiments on two tasks proposed in OpenOOD [41]. The first task involves CIFAR10 as the in-distribution data with CIFAR100 and TinyImageNet (also known as ImageNet200) as out-of-distribution datasets. The second involves TinyImageNet as in-distribution data and SSB [42] (hard split) and NINCO [43] as out-of-distribution data (see the official repository[7] for full details).

We compare our results against the four latest discriminative baselines reported in OpenOOD [41] under the "w/o Extra Data, w/o Training" category, which are KLM [44], VIM [45], KNN [46] and DICE [47]. We report the average AUROC of each task in Table 8. The results are mixed: DiffPath performs the best for the TinyImageNet task, but obtains the poorest result for the CIFAR10 task.

As noted in the main text, to our knowledge near-OOD tasks are not a standard evaluation setup for generative methods. We hypothesize that such tasks are better suited to discriminative methods as gradient-based classification training enables the model to learn fine-grained features specific to each in-distribution class, which we believe is crucial in this context. In contrast, generative models focus on maximizing the likelihood of the overall data distribution and are not explicitly trained to identify subtle discriminative features. This could potentially lead to weaker performance in tasks where the distributions exhibit a high degree of overlap. It should be noted that discriminative methods typically require class labels, while unconditional generative methods do not, thus constraining the use of the former to cases with labelled in-distribution data.

---

[2]https://github.com/openai/improved-diffusion
[3]https://github.com/ahsanMah/msma
[4]https://github.com/marksgraham/ddpm-ood
[5]https://github.com/zhenzhel/lift_map_detect
[6]https://github.com/XavierXiao/Likelihood-Regret
[7]https://github.com/Jingkang50/OpenOOD

