# OpenReview forum: "Out-of-Distribution Detection with a Single Unconditional Diffusion Model"
_NeurIPS.cc/2024/Conference — NeurIPS 2024 poster_

### Official Review · Reviewer_k7Q2 · 2024-07-04

**Soundness:** 2
**Presentation:** 3
**Contribution:** 3
**Rating:** 7
**Confidence:** 4

**Summary:**

This paper proposes an unsupervised anomaly detection method based on diffusion models. The core idea is to leverage the properties of the score function of a pre-trained diffusion model to distinguish samples from different distributions, rather than relying on log-likelihoods or reconstruction error. To this end, the authors first demonstrate that log-likelihoods are not a good metric for differentiating samples from different datasets.  The authors then motivate using the score function, specifically the L2 norm of the score function summed across different time steps as a statistic for differentiating OOD samples. This statistic can be interpreted as the rate of change of diffusion trajectories from the original distribution to the standard Normal distribution. Extending this observation, the authors also incorporate the curvature of the trajectory by considering the derivative of the score function. One key contribution is showing that a single diffusion model trained on a large, diverse dataset, such as ImageNet, can be used for OOD detection across multiple datasets. Experiments are performed on CIFAR-10, CIFAR-100, CelebA, and SVHN datasets, demonstrating higher average AUCROC scores compared to existing methods. Ablation experiments show the effect of various design choices.

**Strengths:**

- One central problem in anomaly detection is that the model is specific to each dataset. This paper proposes a partial solution to this problem, which is of significance to the community.
- The proposed idea of using properties of the diffusion trajectory such as the rate of change and curvature is a novel idea, to the best of my knowledge.
- The paper presents the approach clearly and with sufficient motivation. The writing and organization of the paper are praiseworthy, with the authors first presenting the problem, discussing failure modes of likelihood-based methods, analyzing the diffusion trajectories, and then finally introducing their approach.
- The experimental results are strong, with good performance across datasets. Various ablation studies help understand design choices for the proposed approach.

**Weaknesses:**

There are certain concerns regarding the applicability of the approach to a broader range of problems, and some clarifying questions for the authors.

- The method relies on the availability of a reasonably large and diverse dataset for the domain of interest. For images, ImageNet is an obvious choice, but this raises the question of the applicability of this method to tabular domains, which are especially relevant for scientific and industrial applications.
- The connection to OT in Section 3.4 is valid only if the data distribution is Gaussian, as per my understanding. However, the data distribution, including the image datasets analyzed in this paper, is typically highly multi-modal. This invalidates the connection, and an explanation from the authors would be beneficial.
- It is not clear how the 6D statistic introduced in Section 3.5 is used to distinguish samples since it is not a scalar value that can directly be compared.
- Since a single diffusion model trained on ImageNet is used for the experiment, it is not clear what distinguishes an ID and OOD dataset. What makes a dataset ID in this scenario and what is the difference when evaluating C10 vs SVHN and SVHN vs C10 in Table 3?
- There is at least one other diffusion-based anomaly detection method that leverages properties of the diffusion schedule rather than relying on reconstruction error [1]. A brief discussion on the differences between this approach and the one discussed in the paper would be beneficial.

[1] Livernoche, V., Jain, V., Hezaveh, Y. and Ravanbakhsh, S., On Diffusion Modeling for Anomaly Detection. In *The Twelfth International Conference on Learning Representations*.

**Questions:**

- The results in Table 4 are a bit surprising. Why is a model trained on ImageNet better than a model trained on the ID dataset? For example, when treating C10 as the ID dataset, shouldn’t a diffusion model trained on C10 perform better?
- In this paper, the authors motivate the use of ImageNet due to it being a large and diverse dataset. Do the authors have some thoughts on how to pick such a ‘base’ dataset when applying this method to other domains? Is the idea that this dataset should provide coverage over the ID and OOD datasets?
- The authors provide an explanation for why a higher number of DDIM steps slightly hurt performance, but shouldn’t a smaller time difference make the finite approximation method more accurate?

**Limitations:**

The authors discuss the limitations of their approach in sufficient detail. Some of my points in the weaknesses section echo this discussion.

---

> ### Author Rebuttal · Authors · 2024-08-05
>
> We thank the reviewer for acknowledging the significance of the problem, novelty of our approach, quality of our writing and strong experimental results. Below we provide our response to the reviewer’s concerns and questions.
>
> > method relies on the availability of a reasonably large and diverse dataset...this raises the question of the applicability of this method to tabular domains, which are especially relevant for scientific and industrial applications.
>
> Thank you for raising this point. Our experiments focus on the image domain as diffusion models for images are more mature compared to domains such as tabular data. OOD benchmarks for generative models also mainly focus on images. We believe the general approach proposed in DiffPath would still apply, but experiments would need to bear this out. We will mention tabular data as another domain of future exploration in the conclusion, lines 312-314: "There are several interesting future directions…such as video, audio, language time series and *tabular*…".
>
> > connection to OT in Section 3.4 is valid only if the data distribution is Gaussian. However, the data distribution, including the image datasets analyzed in this paper, is typically highly multi-modal.
>
> We agree that one can only mathematically prove the path is OT if the data is Gaussian, and we acknowledge this in Sec 3.4 (c.f. Line 177-178: “this map is the optimal transport (OT) path if the data distribution is Gaussian”). However, prior work has shown experimentally that for higher-dimensional mixtures and images, the paths match the OT cost up to numerical precision [1]. Hence, we discuss OT as a motivation for DiffPath. We will revise Sec 3.4 to emphasize this further.
>
> > not clear how the 6D statistic introduced in Section 3.5 is used to distinguish samples since it is not a scalar value that can directly be compared.
>
> Thank you for raising this point, which we believe we should make clearer in the paper. For OOD detection, the AUROC computation requires scalar values and as such, we do not use the 6D scores directly. Rather, we fit a GMM to the 6D statistic of the ID training set. During evaluation, to obtain an OOD score for a test sample, we compute the likelihood under the GMM. A lower likelihood implies the sample is “farther” from the ID training set, hence more likely to be OOD. The procedure is shown in line 6 of Algorithm 1.
>
> > it is not clear what distinguishes an ID and OOD dataset. What makes a dataset ID in this scenario and what is the difference when evaluating C10 vs SVHN and SVHN vs C10 in Table 3
>
> As discussed above, the ID dataset is the one whose training set’s statistics (1D or 6D) was fit to a density model (KDE or GMM). At test time, we calculate the likelihood of a sample under the density model to classify whether the sample is ID or OOD. For example, consider the task of C10 vs SVHN. If C10 is ID, we compute the statistics of C10’s training set using the ImageNet diffusion model and fit a density estimator to it, then evaluate using samples from C10 and SVHN’s test sets.
>
> > one other diffusion-based anomaly detection method that leverages properties of the diffusion schedule rather than relying on reconstruction error. A brief discussion on the differences between this approach and the one discussed in the paper would be beneficial.
>
> We thank the reviewer for pointing out [2], which we will discuss in the paper. [2] performs OOD detection by using the distribution of the diffusion time of a noisy test sample. Anomalous samples are farther from the data manifold and have higher diffusion time. The key difference with our approach is that DiffPath computes statistics of the diffusion path rather than diffusion time, and we use a single model while [2] requires either KNN search for each sample, or a parametric model per dataset.
>
> > Why is a model trained on ImageNet better than a model trained on the ID dataset? For example, when treating C10 as the ID dataset, shouldn’t a diffusion model trained on C10 perform better?
>
> We hypothesize that the C10 model is unable to compute the scores accurately for SVHN as the C10 model is unable to generalize to SVHN features. Meanwhile, as ImageNet is diverse, the model has broadly captured the features contained in C10 and SVHN. We note that SVHN, being digits, is not well represented in ImageNet. Yet, it appears the diversity of ImageNet allows the model to generalize beyond the training distribution, hence highlighting the importance of a large and diverse base dataset.
>
> > thoughts on how to pick such a ‘base’ dataset when applying this method to other domains? Is the idea that this dataset should provide coverage over the ID and OOD datasets?
>
> Indeed, our central hypothesis is that the base dataset should be diverse enough such that it broadly covers both ID and OOD. In this work, we chose ImageNet as this was the most readily-available large image diffusion model. Extension to even larger models like Stable Diffusion would serve as interesting future work.
>
> > shouldn’t a smaller time difference make the finite approximation method more accurate?
>
> Apart from what was mentioned in the paper, we hypothesize another reason is we do not compute the full Eq. 9, but only the simple time derivative. As a result, at higher DDIM steps, we may not technically be approaching the true $d \epsilon / d\gamma_t$. However, at the lower steps (50/100) that we use in this work, this approximation does not seem to hinder the performance for OOD detection. We leave the investigation of full JVP computation of the derivative to future work.
>
> Thank you again for your positive review. We hope that we have addressed your concerns. If you have further concerns, please let us know.
>
> [1] Khrulkov, Valentin, et al. "Understanding ddpm latent codes through optimal transport." arXiv preprint arXiv:2202.07477 (2022).
>
> [2] Livernoche, Victor, et al. "On diffusion modeling for anomaly detection." arXiv preprint arXiv:2305.18593 (2023).

---

> > ### Comment · Reviewer_k7Q2 · 2024-08-09
> >
> > Thank you for the clarifications which clear most of my questions.
> > After reading the other reviews and the authors' responses, I think the paper is a useful contribution and should be accepted. I maintain my positive score.

---

> ### Author Response · Authors · 2024-08-10
> **Thank you to the reviewer**
>
> We thank the reviewer for their positive remarks and for acknowledging the contributions of our work!

---

### Official Review · Reviewer_eKbL · 2024-07-04

**Soundness:** 3
**Presentation:** 3
**Contribution:** 3
**Rating:** 7
**Confidence:** 4

**Summary:**

This paper looks at statistics calculated from the path through the diffusion model based on $\int_0^T \| \epsilon_{\phi}(x_t, t) - \epsilon_{\psi}(x_t, t)\|dt$ and use it to discriminate between two distributions $\psi$ and $\phi$. A KDE of the distance of ID scores is used to compute the likelihood of the score of a new test image, and then perform OOD detection.

**Strengths:**

* The paper is well written and easy to follow.
* the idea of computing ID statistics using a network pretrained on a large dataset has proven successful with other techniques (SSL, or classifier trained on Imagenet) so it makes perfect sense to explore this idea with diffusion models.
* The method seems to work better than competitors.
* Interesting link with OT.

**Weaknesses:**

1. The training distribution matters (as stated by the authors themselves), so saying that the DM is an universal OOD detector is a bit of an overclaim.
2. In 3.1, It is stated that "likelihood does not work", but the content does not demonstrate such a strong statement. It only shows that *likelihood obtained directly from a diffusion model does not work as is for OOD detection*. It seems from prior work that likelihood coming from the training of generative models is not suitable for OOD detection -- which is a weaker claim than "likelihood does not work" -- but even this weaker statement is not demonstrated properly since showing it for diffusion models is not sufficient evidence. In addition, the authors end up using a likelihood (that of the KDE) to build their score which makes this wording confusing. This part is nothing more than a motivating experiment (which is, by the way, interesting and fits well in the flow of the paper), so I would not provide such an overinterpretation.

**Questions:**

1. Why use the KDE and not directly the score $\sqrt{\sum_t  \| \epsilon_{\theta} (x_t, t) \|^2_2}$
2. 3.5 why use the first-order term, which disconnects the statistics from OT theory? Why powers are chosen up to three?
3. The method requires 50 FE, but what is the cost of one individual FE? OOD detection applications are often real-time, online or embedded so it is important to detail this point.

**Limitations:**

The authors have adequately addressed the limitations.

---

> ### Author Rebuttal · Authors · 2024-08-05
>
> We thank the reviewer for acknowledging the motivation and importance of extending OOD detection to large pretrained generative models, as well as the quality of our writing. Below we provide our response to the reviewer’s questions and concerns.
>
> > “The training distribution matters (as stated by the authors themselves), so saying that the DM is an universal OOD detector is a bit of an overclaim.”
>
> To clarify, we do not claim our method to be a “universal OOD detector”. Instead, we claim that *with a single model*, our method can perform OOD detection “across diverse tasks” (c.f. Line 6), “that a single model outperforms baselines that necessitates separate models for each distribution” (c.f. Lines 47-49) and that “a generalist model … can also be applied to out-of-distribution detection” (c.f. Lines 310-311).
>
> We were careful to ensure that our claims apply only to the tasks that we have tested in the paper. Certainly, we did not expect DiffPath to be capable of “universal” OOD detection across all domains. We acknowledge this explicitly in the limitations section, where we mention c.f. Line 320-321:  “we consider a DM trained on ImageNet, which may not be general enough for specialized applications such as medical images.”
>
> > "It only shows that likelihood obtained directly from a diffusion model does not work as is for OOD detection. It seems from prior work that likelihood coming from the training of generative models is not suitable for OOD detection -- which is a weaker claim than "likelihood does not work" -- but even this weaker statement is not demonstrated properly since showing it for diffusion models is not sufficient evidence. In addition, the authors end up using a likelihood (that of the KDE) to build their score which makes this wording confusing. This part is nothing more than a motivating experiment (which is, by the way, interesting and fits well in the flow of the paper), so I would not provide such an overinterpretation.”
>
> We thank the reviewer for pointing this out, and for acknowledging the motivating purpose of Sec 3.1. We agree that a more accurate header for Sec 3.1 is that “Diffusion Model Likelihoods Do Not Work for OOD Detection”. We will make this change in the final revision.
>
> > “Why use the KDE and not directly the score”
>
> To calculate the AUROC for OOD detection, one needs to fix a threshold and assign values lower than the threshold as OOD and higher as ID, then integrate over all thresholds. We use a density estimator like KDE because in DiffPath, Theorem 1 does not suggest that OOD samples will have lower score norms than ID samples, only that they are *different*. This is evident in Fig 2,  where all we can say is that the histograms are separated relative to each other. Hence, we fit a likelihood estimator like a KDE or GMM to the ID training samples. This way, OOD samples will have low likelihoods under the estimator, and we use those likelihoods in the AUROC computation. This is a subtle point that we believe is worth discussing in the paper for clarity. We will include this discussion in the final revision.
>
> > “why use the first-order term, which disconnects the statistics from OT theory? Why powers are chosen up to three?”
>
> We are unsure what the reviewer means by “disconnects the statistics from OT theory”. DiffPath 1D considers first-order terms of the Taylor expansion (Eq. 8) while DiffPath 6D uses both first and second-order terms. Both first and second-order terms of the diffusion path are discussed in the OT example in Sec 3.4 (c.f. Lines 187-191: "the corresponding first and second-order OOD statistics are equal and given by $||\frac{dx_i}{dt}||_2 =||\frac{d^2x_i}{dt^2}||_2 = ||a_i e^{-t}||_2...$")
>
> In DiffPath 6D, we combine different powers of the first and second-order terms to form a higher-dimensional statistic. This stems from a purely practical standpoint, as we found that this led to more robust OOD performance. One could in principle investigate other combinations or even higher powers, which we leave to future studies.
>
> > “The method requires 50 FE, but what is the cost of one individual FE? OOD detection applications are often real-time, online or embedded so it is important to detail this point”
>
> Running on a Nvidia A5000 GPU, a single 64x64 sample on our model requires 0.02s (20ms) per FE of the UNet, which we will include in the paper. This is of course hardware dependent. Regardless, due to the nature of diffusion models, we do not expect DiffPath in its current iteration to be suitable for real-time/online OOD detection. This work represents a first step in showing that a single diffusion model can be used for OOD detection (which we believe to be a notable result) and we leave inference speed improvements to future work.
>
> We hope we have resolved your concerns. If so, we kindly ask you to consider raising your score. Should you have any additional issues, please feel free to reach out to us.

---

> > ### Comment · Reviewer_eKbL · 2024-08-08
> >
> > I thank the authors for their response. I increase my rating accordingly.

---

> ### Author Response · Authors · 2024-08-09
> **Thank you to the reviewer**
>
> We thank the reviewer for their positive response and for raising their score!

---

### Official Review · Reviewer_3gUh · 2024-07-07

**Soundness:** 3
**Presentation:** 2
**Contribution:** 3
**Rating:** 6
**Confidence:** 3

**Summary:**

This paper presents a diffusion model trained on a single dataset that can also perform well in OOD detection across diverse tasks. The core concept is Diffusion Paths (DiffPath), which characterizes the properties of the forward diffusion path. Specifically, they measure the rate-of-change and curvature of the diffusion paths, using these as the derivatives of the score and contextualize regarding the optimal transport concept. The method is tested on various benchmarks and generally shows improvement over compared baselines.

**Strengths:**

- The proposed idea of defining and discriminating between ID and OOD datasets using the forward diffusion path is novel.
- The method is comprehensible, and the illustrations are clear.
- This framework shows good experimental results compared to several baselines.

**Weaknesses:**

- In Figure 3, how extensively does the diffusion model need to be trained on a large, diverse dataset?
     - The method is limited by the coverage of the ImageNet-trained diffusion model.
     - Ultimately, to achieve broad coverage, a large-scale pretrained diffusion model is required (e.g. more varisous OOD dataset, medical, manufacturing). Is the contribution of guaranteed generalizability compared to foundational generative models still valid?
     - Add a single trained case that can cover datasets from different domains, not just ImageNet.
- In Table 4, performance drops significantly compared to the baseline in hard-settings like CIFAR10 vs. CIFAR100.
     - The proposed method struggles particularly in distinguishing between ID/OOD datasets with similar semantics.
     - Include experiments and improvement strategies for hard-settings in OpenOOD [1].
- Table 4, add and compare with the latest research baseline, Projection Regret [2].

[1] Zhang, Jingyang, et al. "Openood v1. 5: Enhanced benchmark for out-of-distribution detection." arXiv preprint arXiv:2306.09301 (2023).

[2] Choi, Sungik, et al. "Projection regret: Reducing background bias for novelty detection via diffusion models." Advances in Neural Information Processing Systems 36 (2023): 19230-19245.

**Questions:**

N/A

**Limitations:**

see weakness

---

> ### Author Rebuttal · Authors · 2024-08-05
>
> We thank the reviewer for acknowledging the novelty, clarity and strong experimental results of our work. Below we provide our response to the questions raised by the reviewer.
>
> > In Figure 3, how extensively does the diffusion model need to be trained on a large, diverse dataset?
>
> To clarify, we use the pretrained checkpoint from Improved-DDPM [1] trained on unconditional ImageNet 64x64, which according to the authors [2] is trained for 1.5M steps. We do not retrain/finetune the diffusion model, as the key idea is to leverage existing large pre-trained models. In terms of training duration, we did not perform ablations due to the large compute resources required for ImageNet-scale training. That said, our ablations in Table 4 suggest that a large and diverse base dataset is needed for good OOD performance.
>
> > “The method is limited by the coverage of the ImageNet-trained diffusion model.” ... “to achieve broad coverage, a large-scale pretrained diffusion model is required (e.g. more varisous OOD dataset, medical, manufacturing). Is the contribution of guaranteed generalizability compared to foundational generative models still valid?”
>
> We agree with the reviewer that the performance is dependent on the coverage of the base distribution, in this case ImageNet. We have acknowledged this in the limitations section of the paper, where we hypothesize that the ImageNet model would likely be unsuitable for specialized applications like medical and manufacturing (c.f. Line 320-321:  “we consider a DM trained on ImageNet, which may not be general enough for specialized applications such as medical images.”).
>
> Thus, we do not claim to have a model for OOD detection across *all* domains. Our main contribution is to show that a single unconditional diffusion model can be used for OOD detection across a variety of image tasks which we test in the paper. For specialized applications, one should use a base distribution that provides coverage over such cases, or consider even larger foundation models like Stable Diffusion, which we leave for future studies.
>
> > “Add a single trained case that can cover datasets from different domains, not just ImageNet”
>
> We are unsure what the reviewer means by “a single trained case”. We believe the reviewer is suggesting that we should use a single model other than ImageNet that covers multiple datasets/domains. If so, we unfortunately do not have the computational resources to train such a model from scratch and are unaware of any other existing trained unconditional diffusion model. We would be happy to conduct further experiments if the reviewer can point us to an existing model. We have conducted ablations in Table 4 where the base distributions are CIFAR10, SVHN and CelebA. We find that the performance of these models are inferior to the ImageNet model.
>
> > “In Table 4, performance drops significantly compared to the baseline in hard-settings like CIFAR10 vs. CIFAR100.” and “The proposed method struggles particularly in distinguishing between ID/OOD datasets with similar semantics.” and “Include experiments and improvement strategies for hard-settings in OpenOOD”
>
> We thank the reviewer for pointing OpenOOD out to us. To better study DiffPath's performance on near-OOD tasks, we ran additional experiments based on the benchmark proposed in OpenOOD [1] for C10 and ImageNet200 as in-distribution. We report the average AUROC along with most recent baselines from [1].
>
> |Methods|C10|ImageNet200|
> |:-------------:|:-----:|:-----------:|
> |KLM|0.79|0.808|
> |VIM|0.887|0.787|
> |KNN|0.907|0.816|
> |DICE|0.783|0.818|
> |DiffPath|0.797|0.906|
>
> We see that DiffPath achieves similar performance in C10 to recent methods and achieves the strongest result in ImageNet200. We will include these results in the paper. To further improve near-OOD performance, we believe that incorporation of perceptual features [2] can help; currently, DiffPath works from a KL divergence perspective, which may be less sensitive to small pixel differences. We leave this to future work.
>
> > “Table 4, add and compare with the latest research baseline, Projection Regret”.
>
> We cited PR in related works but did not compare quantitatively as we were not able to find open-source code. For diffusion baselines, we endeavor to test all fairly under the same settings and ID-OOD setups, so we train and evaluate the baselines with the provided code. Given that it is non-trivial to reproduce PR’s implementation without reference code, we opt to discuss PR qualitatively. If the reviewer is aware of code for PR, please let us know and we can run the required experiments.
>
> We hope we have addressed your concerns. If so, we hope that you would consider raising your score. If you have further concerns, please let us know.
>
> [1] https://github.com/openai/improved-diffusion
>
> [2] Nichol, Alexander Quinn, and Prafulla Dhariwal. "Improved denoising diffusion probabilistic models." International conference on machine learning. PMLR, 2021.
>
> [3] Yang, Jingkang, et al. "Openood: Benchmarking generalized out-of-distribution detection." Advances in Neural Information Processing Systems 35 (2022): 32598-32611.
>
> [4] Choi, Sungik, et al. "Projection regret: Reducing background bias for novelty detection via diffusion models." Advances in Neural Information Processing Systems 36 (2023): 19230-19245.

---

> ### Author Response · Authors · 2024-08-13
> **Thank you to the reviewer**
>
> As the discussion period comes to a close, we sincerely thank the reviewer for their helpful comments. We hope that we have addressed the reviewer's concerns in our rebuttal. If the reviewer still has unaddressed concerns, please let us know and we will be happy to discuss further.

---

### Official Review · Reviewer_5CB8 · 2024-07-10

**Soundness:** 2
**Presentation:** 3
**Contribution:** 2
**Rating:** 3
**Confidence:** 3

**Summary:**

The paper proposes DiffPath, an OOD Detection method with foundation diffusion models (i.e., diffusion models over diverse data) that can be applied to any in-distribution dataset. By measuring properties of the diffusion trajectory mapping images to noise, the paper demonstrates some improvements on OOD detection. Notably, the method is faster than most other diffusion-based techniques for OOD detection.

**Strengths:**

- The goal of being able to achieve OOD detection with a foundation diffusion model is important as this can enable modular separation of OOD detection and other downstream tasks.
- The paper is well written, with theoretical connections wherever possible, enabling intuitive understanding and making it easy to follow.
- The experiment in sec 3.5 is interesting and higher-order norms have also been used in the past: for example, check out the Gram Matrix paper [1].

[1] Sastry and Oore. Detecting Out-of-Distribution Examples with Gram Matrices. ICML 2020.

**Weaknesses:**

- Achieves lower performance for near-ood detection tasks.
- The evaluations do not sufficiently demonstrate benefits of a single unconditional diffusion model.  Since two sets of examples are said to be OOD if there is no class-overlap between them and the diffusion model is trained on 1k Imagenet classes, I believe that some of the experiments should be focused on demonstrating ease of constructing OOD detectors for arbitrary sets of imagenet classes as in-distribution or perhaps the entire imagenet-data as in-distribution.
- As a follow-up to the above, it seems like this technique can also be used for learning one-class classification networks. In fact, I feel that the paper should discuss the performance on this task first before considering the natural extension to OOD detection where several classes are in-distribution. It also helps understand the generality of the statistics used to identify OOD examples.
- The use of Theorem 1 (fisher-divergence) to motivate L2-norm of score-function as an indicator for OOD detection is not entirely clear. For example, the fisher-divergence uses difference between score-function outputs estimated over a batch of examples.
- The ablation study with resizing is useful. However, it seems like the diffusion model is trained on 64x64 images and hence, the CIFAR10, CIFAR100 and SVHN images are scaled-up to 64x64 while celeba and textures are scaled down to 64x64. For a fair comparison with methods which directly work at 32x32 resolution,  celeba and textures should be first downsampled to 32x32 and then upsampled to 64x64. Otherwise, ood detection between pixelated CIFAR10 images and comparatively high-res celeba/textures images would not be so meaningful.

**Questions:**

See weaknesses.
1. What hyperparameters do you use for KDE/GMM algorithms? How to select these hyperparameters?
2. Intuitively, why do the trajectory statistics such as norm and rate-of-change vary from class to class?
3. Using an SVHN/CIFAR10/CELEB-A model to directly perform OOD detection was not as successful as using a diffusion model trained on imagenet. However, considering a SVHN diffusion model, if we select one of the SVHN classes as in-distribution, can we achieve effective OOD detection?

**Limitations:**

Yes.

---

> ### Author Rebuttal · Authors · 2024-08-05
>
> We thank the reviewer for acknowledging the strong motivation, insightful writing, and interesting experiments in our paper. Below we provide our responses to the questions raised by the reviewer.
>
> >lower performance for near-ood detection tasks
>
> To better study DiffPath's performance on near-OOD tasks, we ran additional experiments based on the benchmark proposed in OpenOOD [1] for C10 and ImageNet200 as in-distribution. We report the average AUROC along with the most recent baselines from [1].
>
> |Methods|C10|ImageNet200|
> |:-------------:|:-----:|:-----------:|
> |KLM|0.79|0.808|
> |VIM|0.887|0.787|
> |KNN|0.907|0.816|
> |DICE|0.783|0.818|
> |DiffPath|0.797|0.906|
>
> We see that DiffPath achieves similar performance in C10 to recent methods and achieves the strongest result in ImageNet200. We will include these results in the paper. To further improve near-OOD performance, we believe that incorporation of perceptual features [2] can help; currently, DiffPath works from a KL divergence perspective, which may be less sensitive to small pixel differences. We leave this to future work.
>
> >evaluations do not sufficiently demonstrate benefits of a single unconditional diffusion model
>
> Our evaluations show that DiffPath, with a single model, is comparable to baselines trained on in-distribution data. The benefit is therefore the reduction in resources needed to train different models for different tasks. Our work connects OOD detection with recent foundation generative models, where one model excels at multiple tasks. Further, our ablations show that the choice of base distribution matters, and one requires a diverse dataset for DiffPath to work well.
>
> > some of the experiments should be focused on demonstrating ease of constructing OOD detectors for arbitrary sets of imagenet classes as in-distribution... this technique can also be used for learning one-class classification networks.
>
> Thank you for the suggestion. We conducted experiments using single ImageNet64 classes as ID against various OOD classes and report the AUROC for DiffPath-6D.  Due to limited time, we evaluate on a select but diverse set of randomly selected classes. Note that none of the diffusion baselines consider such a task. The results show DiffPath-6D performs well without hyperparameter tuning for this task. This shows DiffPath's potential for one-class classification. We will include these results in the paper.
>
> |ID (below) / OOD (right)|Daisy|Dugong|Altar|Orange|Perfume|
> |:-------------------------:|:-----:|:------:|:-----:|:------:|:-------:|
> |Airship|0.823|0.83|0.818|0.818|0.863|
> |Cheeseburger|0.768|0.86|0.727|0.786|0.958|
> |Pizza|0.824|0.9|0.692|0.842|0.959|
>
> >the fisher-divergence uses difference between score-function outputs estimated over a batch of examples
>
> Thank you for highlighting this distinction, which we will highlight in the revision for clarity. First, we clarify that Theorem 1 serves as motivation and not a guarantee/proof for OOD detection, as stated in line 121 of the paper: “This *motivates* the use of the norm of the score as an OOD statistic…”. The intuition is that if L2 norms of the scores of individual samples are different, so should their expectations (the reverse may not be true). Our experiments and Fig. 2 suggests that this reasoning holds.
>
> >For a fair comparison with methods which directly work at 32x32 resolution, celeba and textures should be first downsampled to 32x32 and then upsampled to 64x64
>
> We thank the reviewer for this suggestion. We agree that standardizing the resolution will allow for fairer comparison. Double resizing introduces twice the interpolation error, hence we retrained baselines at 64x64 resolution so every dataset is resized only once to 64x64. Due to time constraints, we report results for CelebA as ID:
>
> ||CIFAR10|SVHN| CIFAR100|Textures|
> |:-----------:|:-------:|:------:|:--------:|:--------:|
> |MSMA|1|1|1|0.979|
> |DDPM-OOD|0.674| 0.463|0.644|0.841|
> |LMD|0.999|1|0.998|0.98|
> |DiffPath|0.999|1|0.998|0.943|
>
> MSMA, LMD and DiffPath have comparable performance, while DDPM-OOD suffers a performance drop. We will include these updated results and revise our claims to DiffPath ‘matches’ the performance of baselines at 64x64 resolution. Again, we emphasize that DiffPath matches the performance using a single model, while baselines require individually trained models.
>
> >What hyperparameters do you use for KDE/GMM algorithms? How to select these hyperparameters?
>
> We mention the hyperparameter values in appendix B. Selection is done empirically via simple search over a defined set.
>
> >why do the trajectory statistics such as norm and rate-of-change vary from class to class?
>
> As motivated by Theorem 1 and the OT example, our hypothesis is that the path connecting different distributions to standard Gaussian differs. In this work, we observe that the differences manifest in the rate-of-change and curvature of the paths. There could certainly be other measurable properties of the paths that could be explored, which we believe is an exciting future direction.
>
> > considering a SVHN diffusion model, if we select one of the SVHN classes as in-distribution, can we achieve effective OOD detection?
>
> As we are not proposing to use a diffusion model with SVHN as the base distribution, we are unsure of the reviewer’s suggestion. In terms of single-class OOD detection, we have provided results on ImageNet above as suggested.
>
> We hope that we have addressed the reviewer’s concerns. If so, we kindly request to consider raising your score. If you have further concerns, please do not hesitate to let us know.
>
> [1] Yang, Jingkang, et al. "Openood: Benchmarking generalized out-of-distribution detection." Advances in Neural Information Processing Systems 35 (2022): 32598-32611.
>
> [2] Choi, Sungik, et al. "Projection regret: Reducing background bias for novelty detection via diffusion models." Advances in Neural Information Processing Systems 36 (2023): 19230-19245.

---

> ### Comment · Reviewer_5CB8 · 2024-08-08
> **Thank you for your response**
>
> Thank you for your response! I still have a few outstanding concerns/questions:
>
> 1. Open-OOD results: Could you please describe the transformation pipeline (e.g., the sequence of image transformation you applied in order to get the 64x64 images) you used for ImageNet-200?
> 2. One-class learning results: these results seem encouraging! Instead of arbitrary pairs of in-distribution and out-of-distribution classes, it will be more effective to consider semantically related categories for ID/OOD pairs. For example, see Table 8 of [1]. Here, if you consider the dogs subset, you may consider one of them as ID (e.g., beagle) and the remaining 11 dog breeds as OOD. This is a challenging setting while also having important practical applications -- so, demonstrating improvements in this case is going to be significant. I understand that this experiment requires time and compute resources and may be more suitable for a later revision.
> 3. Fair Comparison: Thank you for the additional experiments on this task. Comparing these numbers to the original table, it seems to confirm my speculation that comparing between pixelated images (i.e., upsampling C10/SVHN/C100) and higher-resolution images (e.g., downsampling Celeb-A) leads to over-optimistic results. For e.g., in Table 3, MSMA is shown to yield an AUROC of 0.871 for CelebA vs C10 when using a diffusion model trained at 32x32 resolution (is this correct?) and comparing between original C10 images and downsampled CelebA images. Next, in this new experiment using a diffusion model trained at 64x64 resolution, we notice that it achieves an AUROC of 1.0 and in my understanding, this can attributed to upsampling C10 images to 64x64.  While I understand that DiffPath uses a single model while baseline methods rely upon a separate model in each case, DiffPath also has access to a much stronger diffusion model since its trained on much more data. So, matching the results with the baseline methods is not very surprising; in fact, it also seems to me that MSMA achieves better results and requires fewer function evaluations as compared to DiffPath.
>
> ***
> EDIT: in Q3, I was referring to an experimental setting where a diffusion model is trained on SVHN images and one of the SVHN classes (e.g., 0) is considered as in-distribution and others as OOD. It is similar to the one-class detection results for ImageNet already provided in the above response.  This experiment can evaluate if DiffPath can generalize to a diffusion model trained on much smaller data. Again, this may be more suitable for consideration in a later revision.
>
> [1] Ahmed and Courville. Detecting Semantic Anomalies.

---

> ### Author Response · Authors · 2024-08-09
> **Response to reviewer 1/2**
>
> We thank the reviewer for their timely reply. Below is our response to the reviewer’s comments.
>
> **OpenOOD transformation pipeline**
>
> We use standard transformations in diffusion image modeling: we resize the images to 64x64 using bilinear interpolation, followed by normalization of pixels to the range [-1,1].
>
> **One class learning**
>
> We thank the reviewer for this suggestion. To our knowledge, generative and discriminative OOD methods are evaluated differently in the literature and we are unaware of any generative baselines that are evaluated via the one-class experiments suggested. We agree that such tests would be very challenging for unconditional generative models as they may not learn the "right" features for discriminating the classes.
>
> Due to time constraints, we could only run preliminary trials with select ImageNet dog breeds (Ibizan hound vs bluetick and beagle). Without further tuning, we found that DiffPath did not distinguish the dog breeds. We also tested LMD on the same task and found similar performance. We hypothesize that methods that perform well on this task are likely to be discriminative models which utilize class labels during training, such as those evaluated in [1].
>
> Gradient-based classification training enables the model to learn fine-grained features specific to each dog class, which we believe is crucial in this context. In contrast, generative models focus on maximizing the likelihood of the overall data distribution and are not explicitly trained to identify subtle discriminative features. This could potentially lead to weaker performance in tasks where the distributions exhibit a high degree of overlap.
>
> To enhance the performance of generative methods in this challenging context, one potential improvement could involve augmenting the OOD score calculation with discriminative features [2]. This approach could leverage the strengths of both methodologies: the generative model (e.g., DiffPath) would effectively manage most tasks, while the discriminative features would address more complex cases characterized by high distribution overlap. We leave this to future studies and will include this discussion as a limitation of DiffPath in the revision.
>
> **"DiffPath also has access to a much stronger diffusion model since its trained on much more data. So, matching the results with the baseline methods is not very surprising"**
>
> We respectfully disagree with the reviewer on this statement. We find it surprising that one can perform OOD detection using a generative model that has *not* been trained on ID data or labels, *regardless of the quantity of other data the model is trained on*.
>
> While one could argue that ImageNet provides broad coverage over images like those in CIFAR10 and faces in CelebA, exact data from these distributions are **not** included in ImageNet. Digits of SVHN are even more sparsely covered by ImageNet. Thus, we believe this is a surprising discovery, one which connects OOD detection with recent findings in foundation generative models, where one model excels at multiple tasks. Our work shows that  the curvature statistics of the Imagenet diffusion model captures general properties of images that are useful for OOD detection.
>
> **Fair comparison**
>
> For a more complete comparison with diffusion baselines at 64x64 resolution, we were able to train MSMA and DDPM-OOD for all ID setups. We will include complete results, including LMD, in the final revision (reconstruction for LMD takes $>10^3$ steps per sample, thus we are unable to obtain results at present). We present the averaged AUROC result over all 12 tasks in Table 3 of our paper for simplicity
>
> | Method             | Average |
> |--------------------|---------|
> | MSMA               | 0.951   |
> | DDPM-OOD           | 0.765   |
> | DiffPath           | 0.942   |
>
>
> Upsizing leads to more optimistic results for MSMA (and LMD in earlier experiments), and only marginally better results for DDPM-OOD. In this setting, MSMA and DiffPath are competitive with each other.
>
> In terms of function evaluations (FEs), MSMA is cheaper than DiffPath (10 vs 50) but this isn't a fundamental limitation of DiffPath: MSMA is a discrete noise level NCSN [3], while DiffPath utilizes the continuous-time diffusion formulation with DDIM sampling. DiffPath can be made more efficient by using better diffusion samplers like DPM-Solver [4].
>
> Taken as a whole, the experimental results show strong performance for DiffPath, as it is competitive with baselines using roughly the same order-of-magnitude in FEs (or one/two magnitudes better FE versus DDPM-OOD/LMD), *while using a single model and without significant hyperparameter tuning*. We believe the last point to be most significant, as this work is the first to show this for generative OOD methods.

---

> ### Author Response · Authors · 2024-08-09
> **Response to reviewer 2/2**
>
> We thank the reviewer for suggesting the above experiments that have suggested where improvements can be made to DiffPath. We hope that the reviewer will evaluate our work in the context of our main claims:
>
> - We propose the use of diffusion path statistics, namely the rate-of-change and curvature, for OOD detection.
> - We show that these statistics are obtained from a Taylor expansion about the diffusion process, and can be estimated simply via finite difference of the DDIM sampler.
> - We further show that one can obtain these statistics for different data distributions using a single model, even when the model is not trained on task-specific data.
> - We connect these statistics to score-matching and optimal transport theory, providing a theoretical motivation for DiffPath.
> DiffPath is competitive with generative baselines requiring individually-trained models.
>
> Our experiments were geared towards evaluating these claims. We do **not** claim that DiffPath is the best possible OOD detector in all settings.
>
> [1] Ahmed, Faruk, and Aaron Courville. "Detecting semantic anomalies."
>
> [2] Zhang, Richard, et al. "The unreasonable effectiveness of deep features as a perceptual metric."
>
> [3] Song, Yang, and Stefano Ermon. "Generative modeling by estimating gradients of the data distribution."
>
> [4] Lu, Cheng, et al. "Dpm-solver: A fast ode solver for diffusion probabilistic model sampling in around 10 steps."

---

> > ### Comment · Reviewer_5CB8 · 2024-08-10
> > **Thank you!**
> >
> > I thank the authors for their reply and new insights from additional experiments. I also appreciate the clear explanation of their perspectives. While I strongly subscribe to the claim-based evaluation, I feel that the unconventional experimental settings using both pixelated and non-pixelated images gives the wrong message. More specifically, there are 3 types of comparisons in your experimental setup:
> > 1. **Pixelated vs Pixelated**:
> > - Examples: C10 vs C100, C10 vs SVHN, SVHN vs C10, SVHN vs C100.
> > - Finding: In Table 3, we see that baselines outperform DiffPath in all these cases while DiffPath still offers comparable performance in some cases.
> > 2. **Pixelated vs Non-pixelated**:
> > - Examples: C10 vs Celeb-A, C10 vs Textures, SVHN vs Celeb-A, SVHN vs Textures, Celeb-A vs C10, Celeb-A vs C100, Celeb-A vs SVHN
> > - Finding: In each of these cases, DiffPath achieves close to 100% detection rate. However, this is also very different from the conventional setting and distinguishing between pixelated and non-pixelated images may almost be trivial. In Table 3, DiffPath outperforms all baselines but under fair settings, it seems like all baselines achieve close to 100% detection rate (at least for Celeb-A vs C10/C100/SVHN as in your rebuttal response).
> > 3. **Non-pixelated vs Non-pixelated**:
> > - Examples: Celeb-A vs Textures
> > - Finding: All baselines outperform DiffPath both in Table 3 and in the new Celeb-A results in your rebuttal response.
> >
> > As stated above, my concern is that these are non-conventional settings with over-optimistic results in some cases and cannot be compared directly with other works. A standardized benchmark such as OpenOOD allows us to standardize comparison between methods. It is impressive that DiffPath achieves improvements on Imagenet-200 on the near-OOD tasks in OpenOOD; but the baseline methods in that table operate using 224x224 images while DiffPath operates at 64x64 resolution and hence, it is not clear if it is meaningful to compare between them directly.
> >
> > I would appreciate a standardized comparison between baselines and your method that does not involve pixelated images (i.e., resizing standard 32x32 datasets to 64x64 resolution). This would allow for a fair comparison between DiffPath results and other results in the literature. Finally, while it is not necessary to improve over the baselines in every case, it would be good to demonstrate _some_ improvements over the baseline in the standardized evaluation settings.
> >
> > I request the authors to kindly correct me in case I am misinterpreting the results. Thank you!

---

> > > ### Author Response · Authors · 2024-08-13
> > > **Thank you to the reviewer**
> > >
> > > As the discussion period comes to a close, we sincerely thank the reviewer for their helpful comments and for engaging with us. We hope that we have addressed the reviewer's concerns in our rebuttals and if so, we kindly request that the reviewer consider revising their score. If the reviewer still has unaddressed concerns, please let us know and we will be happy to discuss further.

---

> ### Author Response · Authors · 2024-08-10
> **Response to reviewer**
>
> Thank you to the reviewer for their prompt response and for providing further clarifications. We better understand the concerns raised.
>
> From our interpretation, the reviewer uses the term 'pixelated' to describe images that have been upsampled from a lower resolution (e.g., 32x32 CIFAR10 to 64x64), and 'non-pixelated' to refer to images that have been downsampled from a higher resolution (e.g., CelebA, Textures). We agree with the reviewer’s observation that when comparing 'pixelated' versus 'non-pixelated' images, the blur introduced during upsampling adds additional information—absent in downsampling—that could potentially simplify the OOD task.
>
> To address this concern, we have re-evaluated the results presented in Table 3 of the paper, focusing exclusively on OOD tasks where comparisons are made only between 'pixelated vs pixelated' and 'non-pixelated vs non-pixelated' images. In other words, we have omitted cases where an upsampled image is compared to a non-upsampled image, thereby eliminating any potential advantage due to the blur of upsampling. The revised results are presented below:
>
> | ID           | CIFAR10 | CIFAR10 | SVHN   |  SVHN   | CelebA | Average |
> |--------------|---------|---------|--------|---------|--------|---------|
> | OOD          | SVHN    | CIFAR100| CIFAR10| CIFAR100| Textures|         |
> | Diffusion NLL| 0.091   | 0.521   | 0.99   | 0.992   | 0.809  | 0.681   |
> | Diffusion IC | 0.921   | 0.519   | 0.08   | 0.1     | 0.559  | 0.436   |
> | MSMA         | 0.957   | 0.615   | 0.976  | 0.98    | 0.967  | 0.899   |
> | DDPM-OOD     | 0.39    | 0.536   | 0.951  | 0.945   | 0.773  | 0.719   |
> | LMD          | 0.992   | 0.604   | 0.919  | 0.881   | 0.972  | 0.874   |
> | DiffPath     | 0.92    | 0.593   | 0.924  | 0.936   | 0.946  | 0.864   |
>
> From this revised evaluation, DiffPath remains highly competitive with existing baselines. Specifically, it outperforms DDPM-OOD, NLL, and IC, while closely matching the performance of MSMA and LMD. Again, this is achieved using a single model that is not trained on in-distribution data. We will revise the results to only include comparisons between datasets of the same resolution, e.g., these new results and the new 64x64 results in the rebuttal, as well as moderate our claims accordingly. In light of these findings, along with our methodological contributions, we believe that our work remains novel and represents a significant advancement in the generative OOD literature.
>
> Regarding the reviewer's remarks on the OpenOOD benchmarks, we would like to highlight that in OpenOOD, the near-OOD tasks involve datasets where images vary in size *even within the same dataset* (e.g., NINCO and SSB-hard) [1]. Consequently, there is no standardized approach to ensuring comparisons are exclusively between 'pixelated vs pixelated' and 'non-pixelated vs non-pixelated' images, *even in the original settings tested in OpenOOD*. Therefore, we believe that the focus on resolution within the OpenOOD context may be less critical than anticipated.
>
> [1] https://github.com/Jingkang50/OpenOOD

---

### Author Rebuttal · Authors · 2024-08-05

Thank you to the reviewers for their thoughtful comments and feedback. We are glad that the reviewers found our idea of diffusion paths for OOD detection to be novel, well-motivated and that the paper is well-written.

We find most of the reviewer’s questions revolve around method/experiment clarifications. In general, we would like to reiterate that our main contribution is in proposing statistics of the diffusion path, specifically the rate-of-change and curvature, for OOD detection. We show that these statistics are a natural result of the score-matching formulation of diffusion models, where they are derived from a Taylor expansion of the diffusion process. We further show that the statistics can be obtained from a *single* pre-trained unconditional diffusion model trained on a diverse dataset — this approach contrasts against prior work which focus on training a specific model on the in-distribution data. Our experiments serve to validate this idea and show that DiffPath is competitive with strong *individually-trained* baselines.

Based on the comments, we have made the following revisions:

1. Ran additional near-OOD experiments according to the setup in OpenOOD [1], which show significantly improved performance for DiffPath.
2. Ran additional single-class ImageNet OOD experiments.
3. Ran experiments for baselines at the same resolution for more accurate comparisons, showing DiffPath matches the baselines with only a single model.
4. Several clarifications on the theory and methodology of DiffPath to enhance clarity for the reader.

Please see below for detailed responses to each reviewer.

[1] Yang, Jingkang, et al. "Openood: Benchmarking generalized out-of-distribution detection." Advances in Neural Information Processing Systems 35 (2022): 32598-32611.

---

### Decision · Program_Chairs · 2024-09-25

**Decision:**

Accept (poster)

**Comment:**

The paper proposes a novel approach to OOD detection using a single diffusion model trained on a diverse dataset, leveraging the rate-of-change and curvature of diffusion paths. The reviewers found the core idea innovative and the experiments generally compelling, demonstrating competitive results across various OOD tasks. Reviewer 5CB8 appreciated the theoretical connections and clear presentation but expressed concerns about resolution mismatches and the benefits of using a single model, which the authors addressed through additional experiments, showing that “DiffPath performs competitively with existing baselines” even when accounting for image resolution discrepancies. Reviewer 3gUh noted the method’s novelty and solid results but pointed out limitations related to model generalizability, which the authors acknowledged and positioned as future work. Reviewers eKbL and k7Q2 commended the clear presentation and the significance of extending OOD detection to large pre-trained generative models, with eKbL stating that “the method seems to work better than competitors” and k7Q2 highlighting the value of a solution that reduces the need for task-specific models. The authors’ comprehensive rebuttals effectively addressed the reviewers’ concerns, supporting the recommendation for acceptance due to the paper’s strong methodological contributions and potential impact on the field.